

# Feasibility of Irrigation Monitoring with Cosmic-ray Neutron Sensors

Cosimo Brogi[1], Heye Reemt Bogena[1], Markus Köhli[2], Johan Alexander Huisman[1], Harrie-Jan Hendricks Franssen[1], Olga Dombrowski[1]

[1]Agrosphrum Institute (IBG-3), Forschungszentrum Jülich GmbH, 52425 Jülich, Germany
[2]Physikalisches Institut, Heidelberg University, Heidelberg, Germany

*Correspondence to*: Cosimo Brogi (c.brogi@fz-juelich.de)

**Abstract.** Accurate soil moisture (SM) monitoring is key in irrigation as it can greatly improve water use efficiency. Recently, Cosmic-Ray Neutron Sensors (CRNS) have been recognized as a promising tool in SM monitoring due to their
large footprint of several hectares. CRNS have great potential for irrigation applications, but few studies have investigated whether irrigation monitoring with CRNS is feasible, especially for irrigated fields with a size smaller than the CRNS footprint. Therefore, the aim of this work is to use Monte Carlo simulations to investigate the feasibility of monitoring irrigation with CRNS. This was achieved by simulating irrigation scenarios with different field dimensions (from 0.5 ha to 8 ha) and SM variations between 0.05 and 0.50 $cm_3\ cm^{-3}$. Moreover, the energy dependent response functions of eight
moderators with different high-density polyethylene (HDPE) thickness or additional gadolinium thermal shielding were investigated. It was found that a considerable part of the neutrons that contribute to the CRNS footprint can originate outside an irrigated field, which is a challenge for irrigation monitoring with CRNS. The use of thin HDPE moderators (e.g., 5 mm) generally resulted in a smaller footprint and thus stronger contributions from the irrigated area. However, a thicker 25 mm HDPE moderator with gadolinium shielding improved SM monitoring in irrigated fields due to a higher sensitivity of
neutron counts with changing SM. Such moderator and shielding provided high chances of detecting irrigation events, especially when the initial SM was relatively low. However, it was found that variations in SM outside a small, irrigated field (i.e., 0.5 and 1 ha) can affect the count rate more than SM variations due to irrigation. This suggests the importance of retrieving SM data from the surrounding of a target field to obtain more meaningful information for supporting irrigation management, especially for small irrigated fields.



# 1 Introduction

Soil moisture (SM) affects a variety of biogeochemical and energy-related processes (Vereecken et al., 2008; Seneviratne et al., 2010) and is crucial for understanding hydrological processes in the vadose zone (Brogi et al., 2022). As a reduction in soil moisture availability can negatively affect crop health with repercussions on food security, irrigation is often employed
to prevent yield reduction and crop failure connected to droughts and heat waves (Webber et al., 2016; Siebert et al., 2017; Tack et al., 2017; Kukal and Irmak, 2019; Zaveri and Lobell, 2019). Worldwide, one quarter of the cropped land is irrigated and irrigation accounts for more than 70% of human blue water consumption (Rost et al., 2008). Especially in semiarid and arid regions, irrigation helps to increase food production and stabilize yields (Troy et al., 2015; Kamali et al., 2022). Unfortunately, irrigation can rapidly deplete local freshwater resources. It is thus predicted that water scarcity will be a key
challenge in ensuring food security for the world's growing population, especially in the light of expected climate change (Molden, 2013; Elliott et al., 2014; Pisinaras et al., 2021). To meet this challenge, improvement of water use efficiency in irrigation is crucial (Adeyemi et al., 2017; Abioye et al., 2020). This can be achieved, for example, through an accurate characterization and monitoring of SM in space and time (Vereecken et al., 2008).

A wide range of sensors to monitor spatial and temporal SM variation is available. These are generally either large-scale remote sensing techniques that only sense shallow soil depths (up to 5 cm) and are strongly influenced by vegetation and surface roughness (Walker et al., 2004; Wagner et al., 2007; Bogena et al., 2010), or point-scale instruments that only offer local information (Mohanty et al., 2017). Combining measurements from these two groups is challenging due to the differences in scale and the inherent spatial heterogeneity of SM (Western and Blöschl, 1999; Crow et al., 2012). Recently,
hydrogeophysical methods have led to improvements in the monitoring of SM at larger field-scales (Binley et al., 2015). In particular, Cosmic-Ray Neutron Sensors (CRNS) have been identified as a promising method to close the gap between point- and large-scale measurements of SM (Bogena et al., 2015; Andreasen et al., 2016).

CRNS detect neutrons that are produced by natural cosmic-ray radiation. The detected neutrons are generally in the thermal
(below 0.5 eV) or epithermal (0.5 eV to 0.5 MeV) energy regime. The amount of epithermal neutrons is known to be negatively correlated with the abundance of hydrogen atoms near the soil surface, and thus with the SM nearby the CRNS (Zreda et al., 2008; Desilets et al., 2010; Zreda et al., 2012). CRNS are not only sensitive to SM but also to snow cover (Schattan et al., 2017; Bogena et al., 2020) and, to a generally lesser degree, vegetation (Baatz et al., 2015), atmospheric water vapour (Rosolem et al., 2013), and intercepted water in the canopy and lattice water (Bogena et al., 2013). The
accuracy of CRNS- based SM estimates is affected by environmental conditions. For example, the count rate is inversely proportional to SM such that dry soils have a higher neutron density that allows more accurate measurements compared to wet soils (Desilets et al., 2010; Bogena et al., 2013). Instrument design and set-up can also affect CRNS performance. Thus, recent sensor developments have focused, for example, on enhancing neutron count rates to obtain a higher temporal



resolution for SM estimation, or on refining the sensitivity to thermal or epithermal neutrons (Weimar et al., 2020). Higher

count rates can be achieved with larger sensors or by using multiple counter tubes (Chrisman and Zreda, 2013; Schrön et al., 2017). The sensitivity to neutrons of different energy can be modified by using high-density polyethylene (HDPE) moderators of different size (Desilets et al., 2010; Weimar et al., 2020) as well as by gadolinium- (Ney et al., 2021) or cadmium-based (Andreasen et al., 2016) shielding to prevent the detection of thermal neutrons (Desilets et al., 2010).

A CRNS provides SM information for an area of several tens of hectares and tens of centimetres deep into the soil (Köhli et al., 2015). Compared to point-scale SM monitoring sensors, a CRNS is non-invasive, offers passive and continuous measurements with relatively high temporal resolution, requires low maintenance (Schrön et al., 2018b), and is invariant to environmental variables such as soil temperature (Finkenbiner et al., 2019). In the context of agricultural applications, a CRNS additionally does not present the logistic challenges associated with directly inserted sensors, which need to be

removed and reinstalled during harvest, planting, and other management actions (Franz et al., 2016). CRNS applications have increased rapidly in recent years, even though the principles behind the measurement approach have been known for decades (Kodama et al., 1985). CRNS have been used for the validation of satellite-based remote sensing products (Montzka et al., 2017), the improvement of hydrological model prediction (Shuttleworth et al., 2013) and land-surface model prediction (Baatz et al., 2017), and the monitoring of snowpack dynamics (Schattan et al., 2017; Bogena et al., 2020), among

many other applications. In the upcoming years, additional coverage, real-time data availability, and rover-based measurements are expected to further increase the use of CRNS, for example to study prolonged droughts or flood events (Dong et al., 2014; Jakobi et al., 2020; Bogena et al., 2022).

Cosmic-ray neutron sensing has shown potential for monitoring and informing irrigation (Franz et al., 2020). However, the

most accurate results are obtained in environments where SM within the footprint is rather homogeneous (Schrön et al., 2017). Although Franz et al. (2013) indicated a rather small effect of horizontal SM heterogeneity on CRNS measurements under natural conditions, individual contributions from areas with different SM content can be overlooked by a single CRNS (Schrön et al., 2018a; Schattan et al., 2019; Badiee et al., 2021; Schrön et al., 2022). Sub-footprint heterogeneity can be reconstructed using multiple instruments, but this comes with increased costs and necessitates further assumptions regarding

spatial continuity (Heistermann et al., 2021). As a result, it can be difficult to distinguish local SM variations (Francke et al., 2022), such as the difference between the SM in a small irrigated field and in its surroundings. Furthermore, if the irrigated area is only a small portion of the area sensed by the CRNS, the detection of irrigation-related SM variations might not be possible (Li et al., 2019). Despite such limitations, Ragab et al. (2017) reported that CRNS measurements were useful for monitoring soil moisture deficit in the root-zone and Finkenbiner et al. (2019) found that information obtained from

combined CRNS measurements and electrical conductivity surveys could improve water use efficiency in a field irrigated with a centre pivot system in Nebraska (USA). Also, Baroni et al. (2018) reported a clear response of CRNS to irrigation, although quantification of single irrigation events was not possible due to effects of precipitation and irrigation of nearby





fields. Li et al. (2019) found that it was not possible to accurately monitor drip irrigation with a standard CRNS in a citrus orchard in Spain. This was a consequence of the relatively small area wetted by drip irrigation, which resulted in a small
mean SM change in the instrument footprint. However, it was suggested that better results could be achieved in irrigated fields with a larger wetted area, drier regions, and longer and more intense irrigation periods. These previous studies highlight that it is currently not clear if CRNS can be used as an accurate stand-alone tool in irrigation management. In particular, the effects of the dimension of the irrigated area, SM variation due to irrigation, and the design of the sensor are largely unaddressed.


Within this context, the aim of this study is to analyse the feasibility of CRNS-based SM monitoring in irrigated environments. To achieve this, neutron transport and detection in irrigated environments was investigated with physics-based Monte Carlo simulations, which are widely used in CRNS studies (Köhli et al., 2015; Andreasen et al., 2016). For example, Monte Carlo simulations have been used to describe footprint characteristics of CRNS (Zreda et al., 2008; Köhli et al., 2015)
and to improve local site arrangement and instrument calibration strategies (Desilets and Zreda, 2013; Schrön et al., 2017). In this study, the Ultra Rapid Adaptable Neutron-Only Simulation (URANOS) model developed by Köhli et al. (2015) was used. Simulations were performed for five different dimensions of irrigated areas (i.e., 0.5, 1, 2, 4, and 8 ha) and SM variations between 0.05 $cm^3$ $cm^{-3}$ and 0.50 $cm^3$ $cm^{-3}$ both within and outside the irrigated area. To also evaluate how detector design can help to improve irrigation monitoring, the energy dependent response function of eight different moderators were
also considered. This set of Monte Carlo simulations will allow us to investigate the effect of different moderators, dimensions of the irrigated area, and SM variations inside and outside the irrigated area on CRNS-based irrigation monitoring.

## 2 Materials and methods

### 2.1 Soil moisture monitoring with CRNS

CRNS measure SM by detecting the environmental neutron density produced by cosmic radiation which is inversely related with below- and aboveground hydrogen pools that surround the sensor. These environmental neutrons have different energies depending on the amount and type of interactions that have occurred. Primary cosmic rays with energies around 1 GeV generate high-energy (larger than 20 MeV) neutrons in the atmosphere. By interacting with heavy atoms, these neutrons lose energy and become fast neutrons (0.5 to 20 MeV). The energy of these fast neutrons is further reduced by
elastic collisions with lighter atoms (generally hydrogen), first to the epithermal regime (0.5 eV to 0.5 MeV), and finally to a thermal equilibrium (below 0.5 eV). CRNS typically measure neutrons in the thermal to fast energy regimes. The measured neutron flux is affected by multiple hydrogen pools, such as soil water, water bodies, lattice water, and biomass. Typically, the CRNS signal is mainly controlled by SM variations, but the additional hydrogen pools can strongly influence the accuracy of the SM estimates (Zreda et al., 2012; Baatz et al., 2015; Jakobi et al., 2020; Bogena et al., 2022).






Generally, a CRNS is composed of one or more neutron detectors. These are gas-filled metal tubes that can be bare (thermal-epithermal neutron detection) or moderated with HDPE (epithermal to fast neutron detection). When a neutron from a certain energy range collides with the detector gas, this results in a charge pulse that is registered by the pulse detection module. A data logger periodically retrieves and stores neutron counts and pulse diagnostics from the pulse detection module.

Additionally, a meteorological station can be added for data correction and a modem can be installed to deliver data in near-real time. More detailed information on the main detector components and physics can be found in Zreda et al. (2012), Schrön et al. (2018b), and Weimar et al. (2020).

## 2.2 CRNS footprint and penetration depth

The quantitative description of the horizontal area over which a CRNS measures is named footprint. Detected neutrons that

had no contact with the ground (non-albedo neutrons) are, by definition, excluded from the footprint calculation. Thus, the footprint only depends on detected neutrons that had contact with the ground (albedo neutrons). As the origin of a neutron and whether it had contact with soil cannot be recorded by a CRNS, the footprint is generally obtained via neutron transport simulations. The footprint is assumed to be circular and depends on the Euclidean distance between the points where neutrons had first contact with the ground and the point of detection. A quantile definition is used to define a distance within

which most detected neutrons originate (Zreda et al., 2008; Desilets and Zreda, 2013). Commonly used radii are the one *e*-folding length (~63% of neutrons) and the two *e*-folding length (~86% of neutrons). The footprint varies depending on environmental conditions and instrument characteristics (Schrön et al., 2017). Monte Carlo simulations showed that the two *e*-folding footprint (R86) is ~240 m in fully dry conditions and is reduced by up to 40% with increasing SM and, to a lesser degree, with variations in humidity, vegetation, and other environmental variables (Köhli et al., 2015). The penetration depth

of a CRNS also depends on SM and is higher below the instrument where it ranges between 83 and ~15 cm (Köhli et al., 2015). The characterization of the CRNS support volume in terms of footprint and measurement depth is a complex and ongoing research subject (Schrön et al., 2022), as shown by a range of recent simulation and field studies that further investigated the spatial sensitivity of SM determined with CRNS (Schrön et al., 2017; Badiee et al., 2021; Francke et al., 2022) as well as the footprint of thermal neutrons (Jakobi et al., 2021).

## 2.3 Neutron transport modelling with URANOS

The URANOS model, which is freely available online (http://www.ufz.de/uranos), was used in this study. This model was first developed to address neutron-only interactions and was later adapted to the cosmic-ray neutron problem (Köhli et al., 2015). URANOS is based on a Monte Carlo approach for the simulation of neutron transport and interactions with matter (Köhli et al., 2018). It is tailored to the study of neutron transport in environmental science and thus certain processes such as

gamma cascades or fission are neglected or represented by effective models. This reduces the computational effort and generally allows the simulation of a larger number of neutrons, which results in more accurate simulations (Köhli et al.,

2015). In URANOS, neutrons are emitted from point sources that are randomly distributed within a user-defined source layer with energies sampled from a realistic spectrum matching that of the earth's atmosphere (Sato, 2015). Then, URANOS uses a standard calculation routine that features a ray-casting algorithm for single neutron propagation and tracks the relevant

physical interactions for millions of neutrons (e.g., elastic collisions, inelastic collisions, absorption, and emission processes such as evaporation). URANOS follows the ENDF (Evaluated Nuclear Data File) database standard implementations from Romano and Forget (2013) with cross sections, energy distributions, and angular distributions obtained from the datasets of Chadwick et al. (2011) and Shibata et al. (2011).

## 2.4 Simulation setup

The model domain in URANOS was composed of six layers: one soil and five atmospheric layers. The soil layer extended to 1.6 m depth. The atmospheric layers extended from the soil surface to 1000 m height. The thickness of the five layers was 2, 0.5, 47.5, 30, and 920 m from the bottom to the top layer. The fourth layer was the source layer (from 50 to 80 m above the soil surface, respectively). The pressure of the atmosphere and of the air in the porous media were set to 1020 hPa. A humidity of 3 g cm$^{-3}$ and a composition of 78% nitrogen, 21% oxygen, and 1% argon were assumed. The soil bulk density

was set to 1.43 g cm$^{-3}$ and the porosity was set to 50%.

The simulation domain was 1200x1200 m (144 ha) with a resolution of 1 m. It was divided in two areas: a) a square area at the centre of the domain with five different dimensions (i.e., 0.5, 1, 2, 4, and 8 ha) and b) the surrounding area. The SM in the inner and in the outer areas was modified independently with increments of 0.05 cm$^3$ cm$^{-3}$. The SM variations were

applied homogeneously (both vertically and horizontally) within each area. Figure 1 conceptually describes the simulation strategy taking the 8 ha scenario as an example. A first simulation is performed with a uniform SM of 0.05 cm$^3$ cm$^{-3}$ in both inner and outer areas. Then, the SM is varied either in the inner area, the outer area, or in both areas. This results in 100 simulations with different SM combinations for each scenario, and a total of 500 simulations. For each simulation, 10$^8$ neutrons were used as this provided sufficient precision and a reasonable computation time (Köhli et al., 2015).



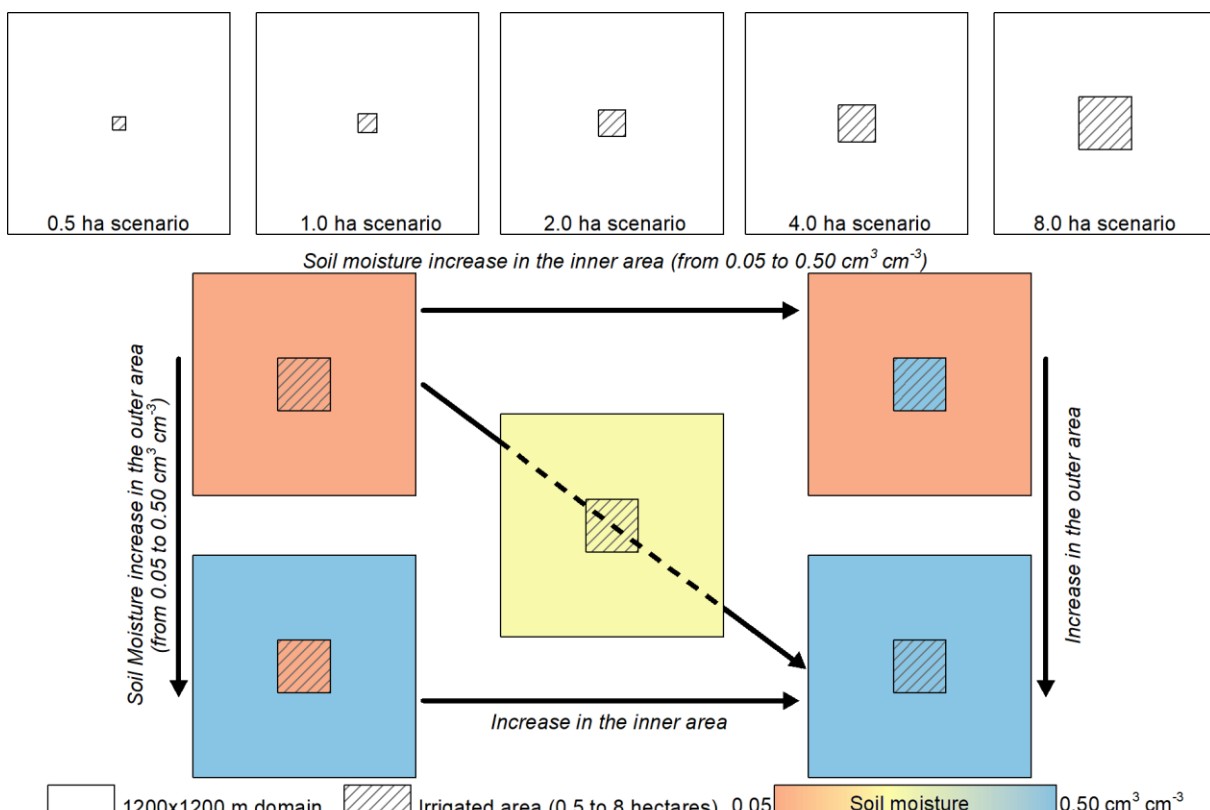

**Figure 1: Examples of the dimensions of the inner area relative to the simulation domain and schematization of the simulation design and setup with an example for the 8 ha scenario. A 1200x1200 m domain is used with an inner area of 8 hectares. The SM is then systematically varied in the inner and outer area with increments of 0.05 cm³ cm⁻³ for a total of 100 simulations.**

## 2.5 Investigated moderators and their response efficiencies

Each simulation provided information on the simulated neutrons that crossed the detector at the centre of the domain (e.g., energy at detection, coordinates of first soil contact). The detector was a vertical cylinder of 9 m radius positioned in the second atmospheric layer (2 to 2.5 m aboveground). However, not all neutrons that pass through the detector tube are detected, whereby the probability of detection depends on neutron energy and direction, as well as on sensor characteristics such as the used conversion gas, geometrical configuration, and moderator type (Köhli et al., 2018). Different moderators are commercially available and are typically made of HDPE of various thicknesses. In this study, we investigated the use of 5, 10, 15, 20, 25, 30, and 35 mm HDPE moderators. Furthermore, a moderator composed of 25 mm HDPE and an additional gadolinium oxide ($Gd_2O_3$) shield was investigated. The energy dependent response functions of the investigated moderators as reported by Köhli et al. (2018) are shown in Figure 2. Detectors with thin HDPE moderators (e.g., 5 mm HDPE) are highly sensitive to thermal neutrons while detectors with thicker moderators have a higher sensitivity to epithermal neutrons. The 25 mm HDPE moderator with gadolinium shielding has similar sensitivity to epithermal neutrons compared to the non-shielded variant but less sensitivity to thermal neutrons. In fact, such shielding can absorb and thus prevent the detection of

more than 90% of the incoming thermal neutrons (Ney et al., 2021), which have a smaller footprint and different sensitivity to SM than epithermal neutrons (Jakobi et al., 2021). A cubic spline of each response function was applied to the output of

each simulation and a weight was assigned to each neutron depending on its energy at detection. Then, these weights were either summed to obtain the number of detected neutrons or used in a weighted calculation of the R86. The effect of the angular distribution of the neutrons was not considered here as it was assumed to be negligible due to the absence of changes (e.g., vegetation, atmospheric pressure, humidity) in the vicinity of the detector between simulations.

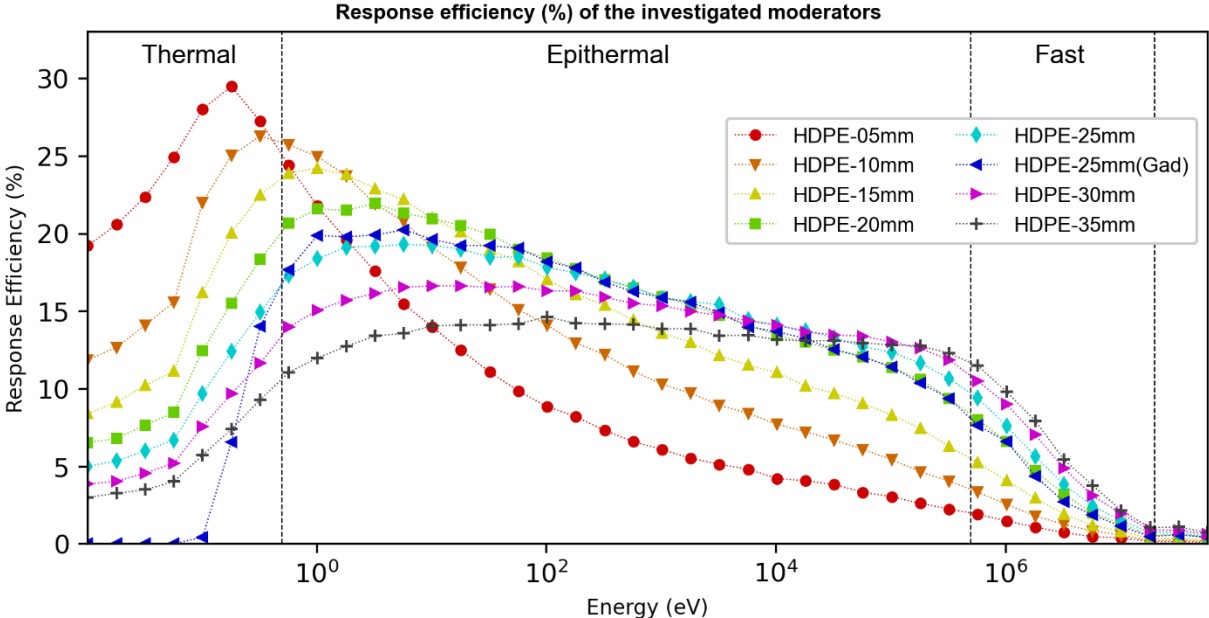


**Figure 2: Energy dependent response functions of different moderators made of 5 to 35 mm thick high-density polyethylene (HDPE) and, in the case of 25mm HDPE, an additional gadolinium based thermal shielding.**

**2.6 Investigation of the feasibility of irrigation monitoring with CRNS**

The detected neutrons in each simulation were divided in two types: albedo and non-albedo neutrons. Albedo neutrons carry

environmental SM information and were further divided in neutrons that originated within the inner area and neutrons that originated in the outer area. Here, it was assumed that the neutrons that originate within the inner irrigated field carry the bulk of the information of interest. Next, the changes of R86 due to SM variations in the inner and outer areas as well as the use of different moderators were analysed. Although it has been shown that a footprint description that relies on a single value can be misleading (Badiee et al., 2021), the R86 represents a standard in CRNS applications and was thus selected to

investigate the simulations of this study. Here, it is assumed that having a relatively small R86 is beneficial when monitoring irrigation in small fields. In a following step, the relative changes in detected neutrons with SM variations were investigated. For a given dimension of the inner area, moderator type, and SM of the outer area (10 simulations), the highest simulated neutron count was set to 100%. Then, the results of the other nine simulations were scaled to that count rate. The influence


of the moderator type, the dimension of the irrigated area, and the SM in the outer area were then compared. Here, larger changes of detected neutrons were considered beneficial as this leads to higher accuracy of the SM estimates.

Next, the sensitivity to irrigation events was assessed in more detail for the moderator type that provided the largest relative changes in detected neutrons. Starting from homogeneous SM conditions in the simulated domain, an irrigation event was assumed to increase the SM in the irrigated area either by 0.05 or by 0.10 $cm^3cm^{-3}$. The initial homogeneous SM conditions were between 0.05 and 0.25 $cm^3cm^{-3}$ as this is the relevant range for irrigation applications. The selected detection thresholds, shown in Table 1, were based on the relative error of the simulations:

$$\sigma = 100 * \sqrt{1/N_1 + 1/N_2} \tag{1}$$

where $N_1$ and $N_2$ are the neutron count obtained with the initial and with the final SM conditions, respectively. In addition to $\sigma$, a value of $\alpha = 1\%$ was included in each threshold to represent a generic detection uncertainty limit for a detector that can achieve ~1000 counts per hour and aggregation times of < 12 hours that are relevant in irrigation monitoring (Schrön et al., 2022).

**Table 1: Detection thresholds that were adopted to investigate the CRNS sensitivity to irrigation events.**

| Threshold value | Sensitivity |
| --- | --- |
| Below $\sigma + \alpha$ | Uncertain results |
| $\sigma + \alpha$ to $2\sigma + \alpha$ | Good detection chance |
| $2\sigma + \alpha$ to $3\sigma + \alpha$ | High detection chance |
| Above $3\sigma + \alpha$ | Detectable |

Lastly, relative changes in detected neutrons due to homogeneous SM variations within the simulation domain were compared with those due to SM variations that occur in the inner or in the outer areas only. Again, the moderator type that had provided the largest relative changes in detected neutrons was selected. For each dimension of the irrigated area (100 simulations), the highest simulated neutron count was set to 100%. The reduced sensitivity of the detection-SM relationship in the case of SM variations that occur only in the inner area was assessed. Here, a lower reduction in sensitivity was considered beneficial for irrigation monitoring. Also, the influence of the SM in the outer area on the count rate was compared to that of SM variations in the inner area. A strong influence of the SM in the outer area was considered disadvantageous for irrigation monitoring.





## 3 Results and discussion

### 3.1 Detected albedo neutrons and their origin

For each dimension of the simulated inner area and for each moderator type, Table 2 shows the minimum and maximum percentages of detected neutrons that originate in the inner area depending on SM conditions. This percentage represents the

detected neutrons that originate within an irrigated field and thus carry the information of interest in case of irrigation applications. Considerable differences are found between simulations depending on the SM in the inner and outer area. The dimension of the irrigated area obviously also influences the results. In particular, 27.7 % to 61.1 % of the detected neutrons originate from within a 0.5 ha irrigated field whereas larger fields show higher percentages (e.g., 51% to 85% from within an 8 ha field). Also, thinner moderators generally show a higher percentage of detected neutrons from the inner area.


**Table 2: Minimum and maximum percentage of detected albedo neutrons originating within the inner area depending on SM conditions with different moderators and different scenarios. The colour in each cell indicates the mean percentage with low values in white and high values in grey.**

| | Scenario (inner area dimension) | | | | |
| --- | --- | --- | --- | --- | --- |
| **Moderator** | **0.5 ha** | **1 ha** | **2 ha** | **4 ha** | **8 ha** |
| **5 mm** | 34.5 - 61.1 | 41.0 - 67.9 | 48.1 - 75.1 | 57.7 - 81.4 | 66.4 - 85.3 |
| **10 mm** | 32.0 - 59.9 | 38.3 - 66.1 | 44.7 - 72.8 | 53.9 - 78.9 | 62.3 - 83.0 |
| **15 mm** | 30.5 - 59.0 | 36.5 - 64.9 | 42.4 - 71.2 | 51.1 - 77.0 | 59.2 - 81.2 |
| **20 mm** | 29.4 - 58.1 | 34.9 - 63.6 | 40.4 - 69.4 | 48.5 - 75.0 | 56.1 - 79.2 |
| **25 mm** | 28.8 - 57.6 | 34.1 - 62.9 | 39.3 - 68.5 | 47.0 - 73.9 | 54.4 - 78.0 |
| **25 mm Gad** | 27.3 - 57.3 | 32.6 - 62.5 | 37.5 - 68.1 | 45.1 - 73.5 | 52.4 - 77.9 |
| **30 mm** | 28.2 - 57.0 | 33.3 - 61.9 | 38.2 - 67.3 | 45.5 - 72.6 | 52.4 - 76.6 |
| **35 mm** | 27.7 - 56.4 | 32.5 - 61.1 | 37.2 - 66.3 | 44.1 - 71.3 | 50.7 - 75.3 |

Figure 3 shows the variation of the percentage of detected neutrons that originate in the inner (Figure 3a) or in the outer area (Figure 3b) for the 0.5 ha scenario as a function of the SM of both areas for a detector with a 25 mm HDPE moderator with gadolinium shielding. For a more detailed description of the detected non-albedo neutrons, please refer to Appendix A. Clearly, the fraction of neutrons originating in the inner and outer area strongly depends on SM. The percentage of neutrons originating in the inner area is smallest when the inner area is wet and the outer area is dry (27.3%). This percentage

increases up to 45.3% if the SM in the inner area is reduced to 0.05 cm$^3$ cm$^{-3}$ and then to 57.3% when the SM of the outer area increases to 0.50 cm$^3$ cm$^{-3}$. The percentage of neutrons originating in the outer area (Figure 3b) shows the opposite trend. From these results, it is clear why CRNS applications focused on irrigation of small fields can face considerable challenges. Especially in small irrigated fields, the number of detected neutrons that originate outside the irrigated area can be higher than the number of neutrons that originate inside of the irrigated area. This is especially true when the SM outside

the irrigated area is relatively low, which is often the case when the inner area is irrigated. Figure 3 also presents the same





analysis for an 8 ha field and a detector with 25 mm HDPE moderator with gadolinium shielding. The percentage of neutrons originating from the inner (Figure 3c) and outer (Figure 3d) area varies with a similar pattern as in the 0.5 ha case. However, the percentage of neutrons from the inner area is higher in the 0.8 ha scenario and shows a lower overall variation. Again, the lowest percentage of detected neutrons originating from the inner area (52.4%) is found when the inner area is

wet and the outer area is dry, whereas the highest value (77.9%) is found with reversed SM conditions. A comparison of the 0.5 ha scenario (Figure 3a-b) and the 8 ha scenario (Figure 3c-d) shows a clear difference in the sensitivity to SM variations. The 8 ha scenario shows higher sensitivity to SM variations in the inner area and less sensitivity to SM variations in the outer area compared to the 0.5 ha scenario.

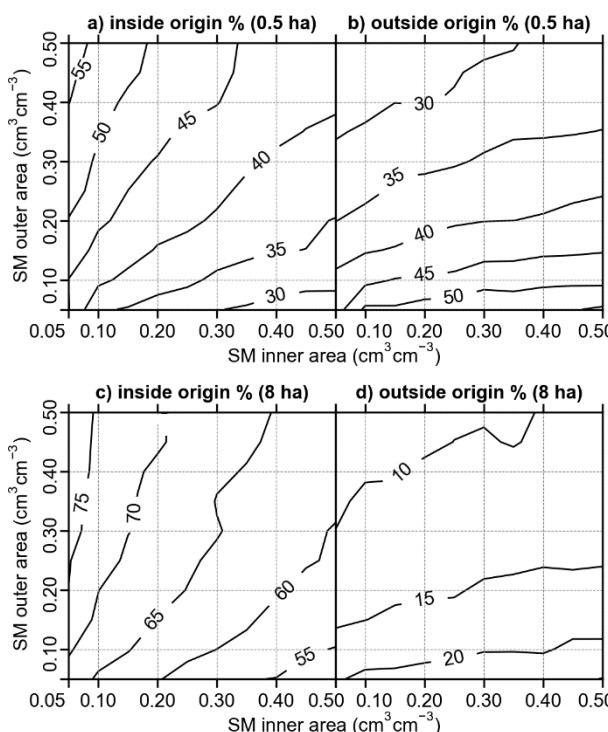


**Figure 3: Percentages of detected neutrons that originate in the a) inner and b) outer area for the 0.5 ha scenario or in the d) inner and e) outer area for the 8 ha scenario. These results are shown for a detector with a 25mm HDPE moderator with gadolinium shielding.**

**3.2 CRNS footprint variations with SM heterogeneity and moderator type**

The analysis of the CRNS footprint is useful for understanding the area from which the measured neutrons originate. The boxplots of R86 for all moderators and for different dimensions of the irrigated field are shown in Figure 4. In general, the footprint increases when a thicker HDPE moderator is used and increases further when gadolinium shielding is added. Additionally, there is a trend towards a larger variability of the footprint size with SM variations when larger inner areas are considered. The largest R86 values are obtained using a 25 mm HDPE moderator with gadolinium shielding (min =147, med





=196, and max =273 m in the 8 ha scenario), whereas the lowest values are obtained with a 5 mm HDPE moderator (min =127, med =162, and max =238 m in the 0.5 ha scenario).

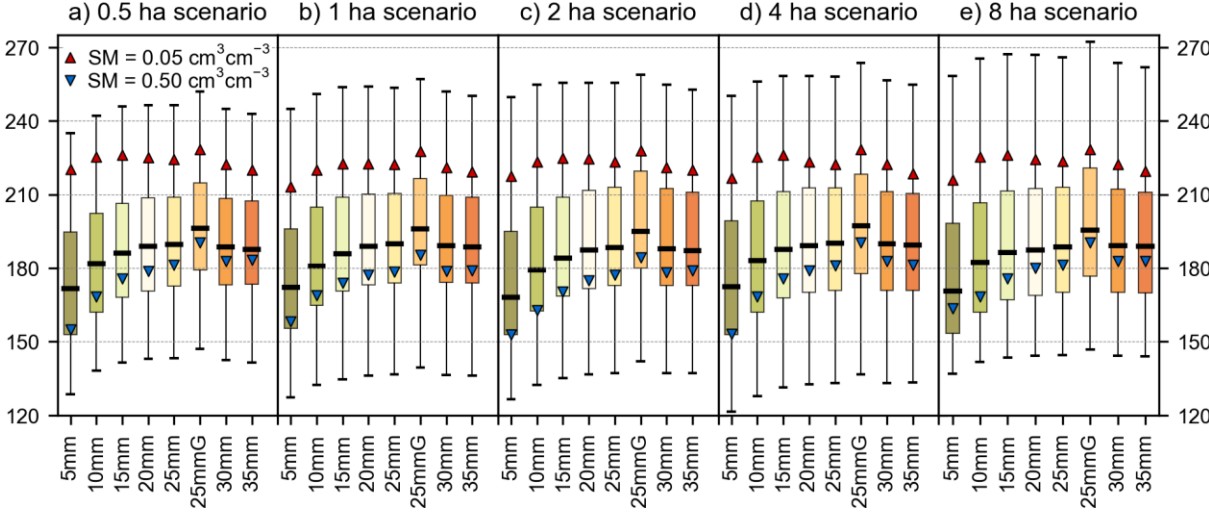

**Figure 4: Boxplots of R86 in meters for different moderator types and size of the inner area, i.e. a) 0.5 ha, b) 1 ha, c) 2 ha, d) 4 ha,**
**and e) 8 ha. Each boxplot refers to one moderator type (100 combinations of SM) and shows the minimum and maximum values with whisker caps, the interquartile range with bars, and the median with a black line. Blue and red triangles show R86 values for homogeneous SM of 0.05 and 0.50 cm$^3$ cm$^{-3}$ respectively.**

A more in-depth analysis of how R86 changes with SM variation is shown in Figure 5 for a 25 mm HDPE moderator with gadolinium shielding. The value of R86 depends on both the SM of the inner and outer area. Similar patterns in R86

variation with SM are found when other moderators are used (not shown). The highest R86 value in the 0.5 ha scenario is ~250 m for high SM in the inner and low SM in the outer area (bottom right in Figure 5a). The R86 decreases to ~229 m when the SM in the inner area is reduced to 0.05 cm$^3$ cm$^{-3}$. A more pronounced decrease of R86 to ~147 m occurs when the SM in the outer area is increased up to 0.50 cm$^3$ cm$^{-3}$. When the size of the inner area increases (Figure 5b-e), the general trends in R86 with SM remain rather constant except for simulations where SM is high in the inner area and low in the outer

area. For these conditions, R86 tends to increase with increasing size of the inner area. For example, R86 is ~271 m in the 8 ha scenario (Figure 5e), which is more than 20 m larger than the R86 in the 0.5 ha scenario (~250 m) for a SM of 0.50 cm$^3$ cm$^{-3}$ in the inner area and 0.05 cm$^3$ cm$^{-3}$ in the outer area.

In previous studies, the smallest R86 was often assumed to occur for the highest SM in the CRNS surroundings. This is true

for homogeneous SM conditions, but may be different for heterogeneous SM distributions (Schrön et al., 2022). When the SM of the inner area is low and that of the outer area is high, the outer area becomes a less important source of neutrons and thus the footprint is reduced. An opposite effect occurs when the inner SM is high, and the outer SM is low. In this case, the inner area is a smaller source, and the neutrons from the outer area become more important, resulting in a larger footprint.

For example, the R86 of a 1 ha field reduces by 38.7 % when SM increases from 0.05 to 0.50 cm$^3$ cm$^{-3}$ in the outer area

whereas the same SM increase in the inner area enlarges R86 by 11.2 %.

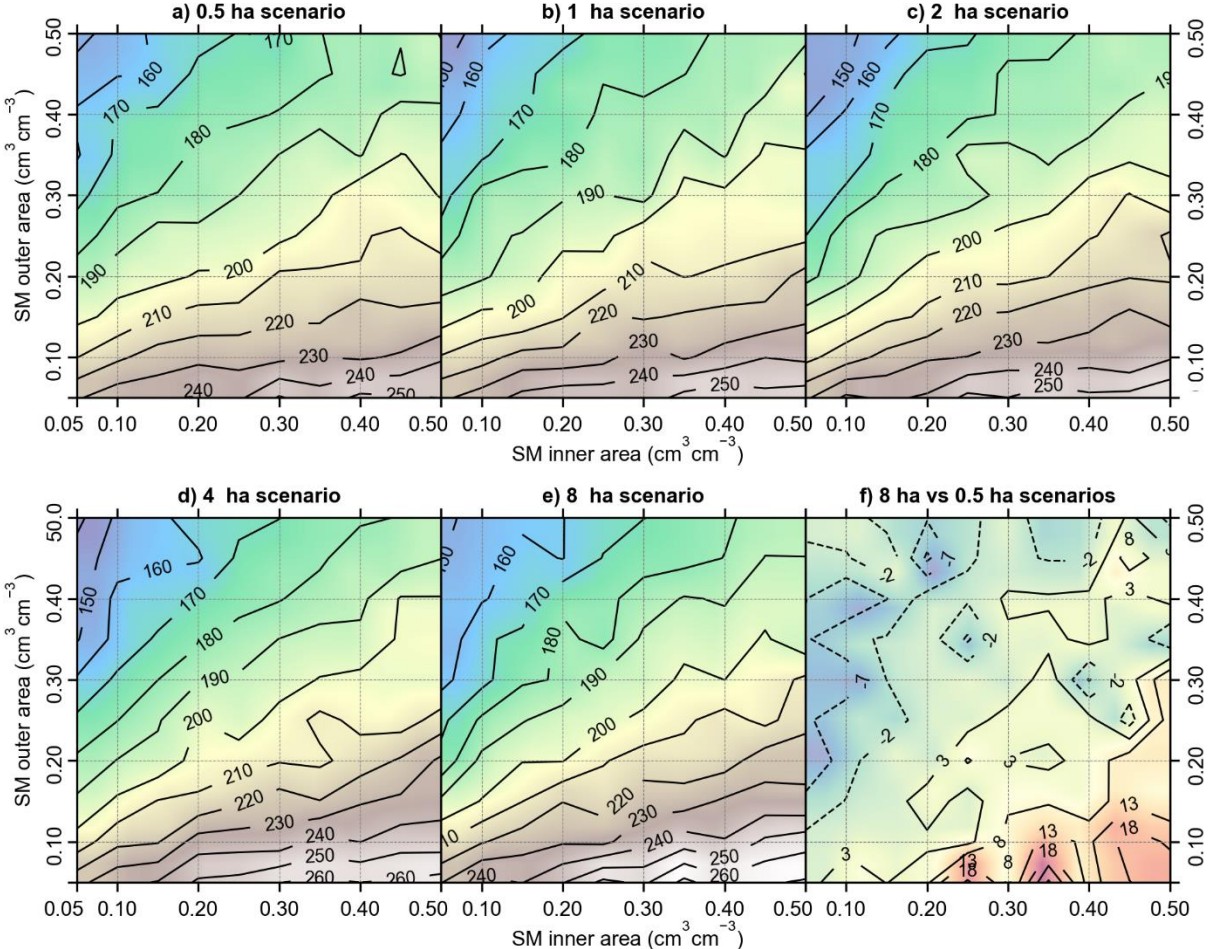

**Figure 5: Variation of R86 with SM using a 25 mm HDPE moderator with gadolinium shielding for the a) 0.5 ha, b) 1 ha, c) 2 ha, d) 4 ha, and e) 8 ha scenario. The differences between the 8 ha and 0.5 ha scenarios are shown in f). Here, negative values indicate**
**that the footprint of the 8 ha scenario is smaller than that of the 0.5 ha and vice versa.**

### 3.3 Effect of spatial heterogeneity on the relationship between neutron counts and SM

The relationship between neutron count rate and SM was mostly investigated for homogeneous SM conditions. Here, we

also explore the effect of SM variations that occur only within an inner area that surrounds the instrument (e.g., through

irrigation measures). We considered sizes of the inner irrigated area from 0.5 to 8 ha and two opposite wetness situations for

the outer area (i.e., SM of 0.05 and 0.50 cm$^3$ cm$^{-3}$). In addition, we considered the effect of the thickness of the moderator

and additional shielding on the number of detected neutrons. In Figure 6 and in Figure 7, the neutron counts of the different

simulations are scaled to the case with the highest count rates (SM of 0.05 cm$^3$ cm$^{-3}$ in the simulated domain). Generally,

neutron count rates show a non-linear negative relationship with increasing SM in the irrigated area. The detector with a 5
mm HDPE moderator shows the smallest relative changes in neutron counts (Figure 6). Thicker HDPE moderators (i.e., 15,

25, and 35 mm) result in larger relative changes. The largest difference in relative changes is observed between a 5 mm and
15 mm HDPE moderators whereas the differences between a 25 mm and a 35 mm HDPE moderator are rather small. The
highest relative change in number of detected neutrons occurs for a 25 mm HDPE moderator with gadolinium shield. This
seems to contradict the large R86 values found for the 25 mm HDPE moderator with gadolinium shielding and is likely due
to the low amount of detected thermal neutrons (< 5%), that only contain limited SM information (see Appendix B). The SM

335  in the outer area also has an influence on the results, although to a lesser degree. For example, a SM of 0.50 cm$^3$ cm$^{-3}$ in the
outer area (Figure 6b) leads to greater variations in detected neutrons than a SM of 0.05 cm$^3$ cm$^{-3}$ in the outer area (Figure
6a).

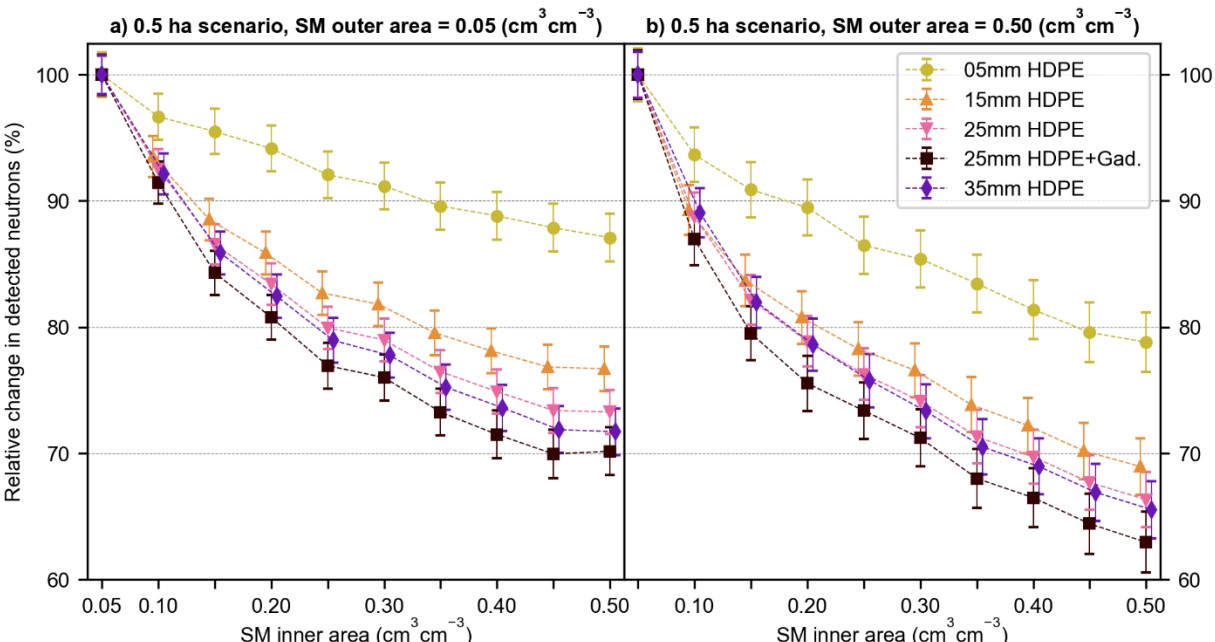

340  **Figure 6: Relative change in number of detected neutrons due to SM variations obtained using different moderators (i.e., 5, 15, 25,
35 mm HDPE and 25 mm HDPE with gadolinium shielding) for a) 0.05 cm$^3$ cm$^{-3}$ SM and b) 0.50 cm$^3$ cm$^{-3}$ SM in the outer area of
a 0.5 ha field. The error bars indicate the relative error of the simulations.**

The neutron count rates obtained with a 25 mm HDPE moderator with gadolinium shielding for sizes of the irrigated area
from 0.5 to 8 ha are shown in Figure 7. An increased size of the irrigated area results in larger relative changes in the number

345  of detected neutrons with SM variations, especially when the outer area is dry (Figure 7a). Higher SM of 0.50 cm$^3$ cm$^{-3}$ in
the outer area (Figure 7b) results in larger relative changes in the number of detected neutrons, although this is mostly the
case for relatively small dimensions of the irrigated area. For the 8 ha scenario, the impact of the SM in the outer area is very
limited.



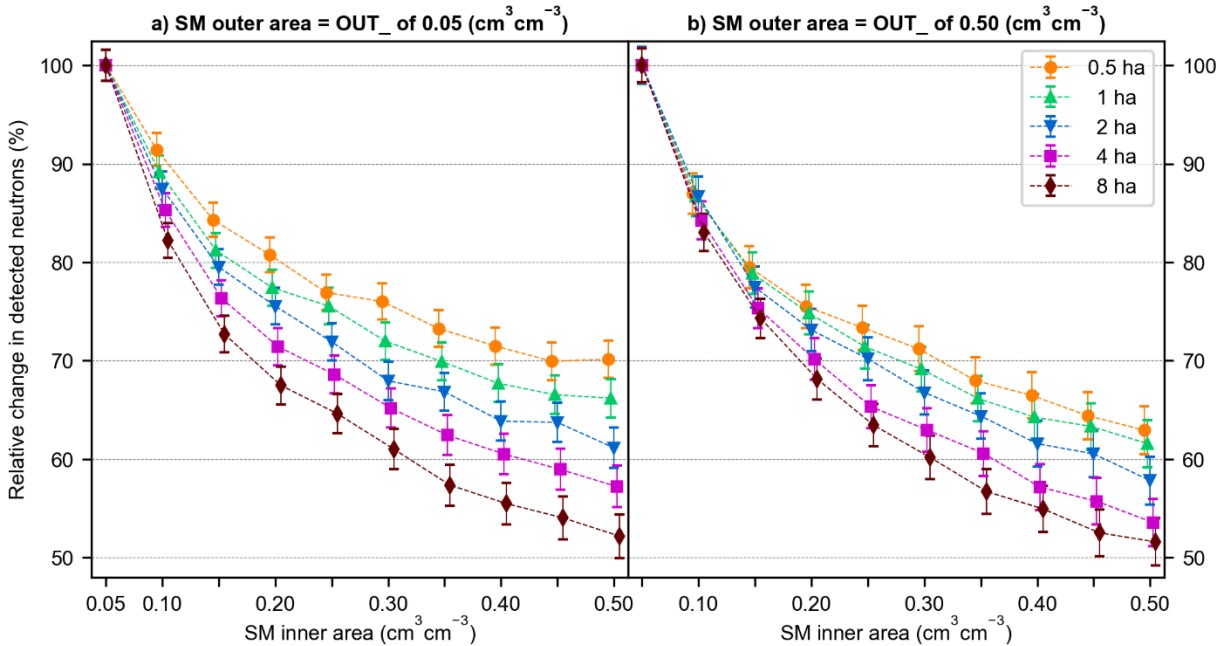

**Figure 7: Relative change in number of detected neutrons using a 25 mm HDPE moderator with gadolinium shielding for different dimensions of the irrigated area and two different SM in the outer area: a) 0.05 cm³ cm⁻³ and b) 0.50 cm³ cm⁻³. The error bars indicate the relative error of the simulations.**

Generally, larger relative changes in detected neutrons with SM variations lead to an improved performance of a given CRNS in a certain environment. Thus, our analysis shows that the CRNS performance in an irrigated environment depends on both the moderator type and the SM outside of the irrigated area. However, the effect of different moderator types on the performance is stronger. A relatively thin HDPE moderator will generally result in low changes of the neutron count rate with SM variations, independent from the dimension of the irrigated field. A thicker HDPE moderator will provide higher neutron count changes and thus better sensitivity to SM changes in the inner area. Overall, a 25 mm HDPE moderator with gadolinium thermal shielding achieve the best results.

### 3.4 CRNS sensitivity to irrigation events

The sensitivity of a CRNS is further analyzed to assess if it is possible to detect irrigation events in fields between 0.5 and 8 ha in size (Figure 8). A 25mm HDPE moderator with gadolinium shielding is used as it provides larger relative changes in the neutron count rate compared to the other investigated moderators. An irrigation event is assumed to increase the SM of the irrigated area by 0.05 or by 0.10 cm³cm⁻³ starting from a homogeneous SM condition within the domain. The detection thresholds (see Figure 8) are based on the relative error ($\sigma$) of the simulations (Eq. 1), which varied between 2.3 and 3.4 % depending on the number of detected neutrons.



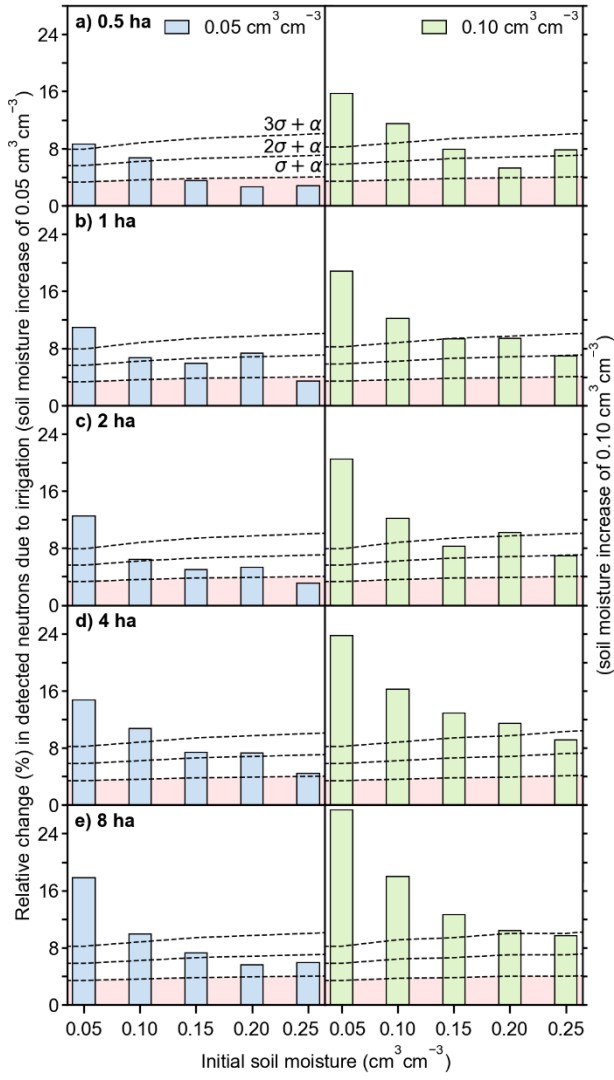

**Figure 8: CRNS chance of detecting irrigation events of 0.05 and 0.10 cm³ cm⁻³ (blue and green bars respectively) in a) 0.5 ha, b) 1 ha, c) 2 ha, d) 4 ha, and e) 8 ha irrigated fields. The bars show the relative change in detected neutrons induced by the irrigation event while the dashed lines show the prescribed detection thresholds. The red area below the $\sigma + \alpha$ threshold indicates uncertain detection.**

As shown in Figure 8, an irrigation event that leads to a 0.05 cm³cm⁻³ increase in SM can be detected with CRNS (relative change in detected neutrons higher than $3\sigma + \alpha$) when the initial SM of the simulation domain is 0.05 cm³cm⁻³. This is the case for all five investigated dimensions of the irrigated area. In the case of 4 ha and 8 ha fields, detection of this type of irrigation event is also achieved with an initial SM of 0.10 cm³ cm⁻³. For values of initial SM larger than 0.05 cm³ cm⁻³, the chance of detecting irrigation events depends on the dimension of the irrigated area with larger irrigated fields yielding higher detection chances. Overall, the detection of a 0.05 cm³cm⁻³ irrigation event is uncertain (relative change in detected neutrons lower than $\sigma + \alpha$) when the initial SM is 0.15 cm³cm⁻³ or higher in a 0.5 ha field (Figure 8a) or 0.25 cm³cm⁻³ in 1





ha and 2 ha fields (Figure 8b-c). Irrigation events that induce larger SM variations of 0.10 $cm^3cm^{-3}$ result in larger relative changes in detected neutrons and thus have a higher chance of detection. Here, the CRNS detects irrigation events with initial SM up to 0.10 $cm^3cm^{-3}$ in 0.5, 1, and 2 ha fields, and with initial SM up to 0.20 $cm^3$ $cm^{-3}$ in 4 ha and 8 ha fields. If the investigation of a 0.10 $cm^3cm^{-3}$ irrigation event is extended to higher initial SM (i.e., 0.30 to 0.40 $cm^3$ $cm^{-3}$), the only

uncertain detection (lower than $\sigma + \alpha$) is that of a starting SM of 0.40 $cm^3$ $cm^{-3}$ in a 0.5 ha field (not shown).

### 3.5 Influence of SM in the outer area

In fields between 0.5 ha and 8 ha, a CRNS is not only sensitive to the irrigation of the target field but also to SM variations in the outer area (e.g., precipitation events or irrigation of neighboring fields). It is thus important to assess the impact of SM variations that occur outside the field of interest. Figure 9 shows the relationships between SM variations (that occur in the

inner area, in the outer area, or in both) and the consequent relative change in detected neutrons. SM variations that are limited to the irrigated field show a reduced sensitivity of the relative change in neutron count rate compared to the case where SM variations occur homogeneously in the entire domain, which is due to the influence of the SM value in the outer area. The strongest reduction in the sensitivity of this relationship is found for the 0.5 ha field (Figure 9a). The sensitivity increases with larger dimensions of the irrigated area and the observed reduction is lowest in the case of the 8 ha field

(Figure 9e).

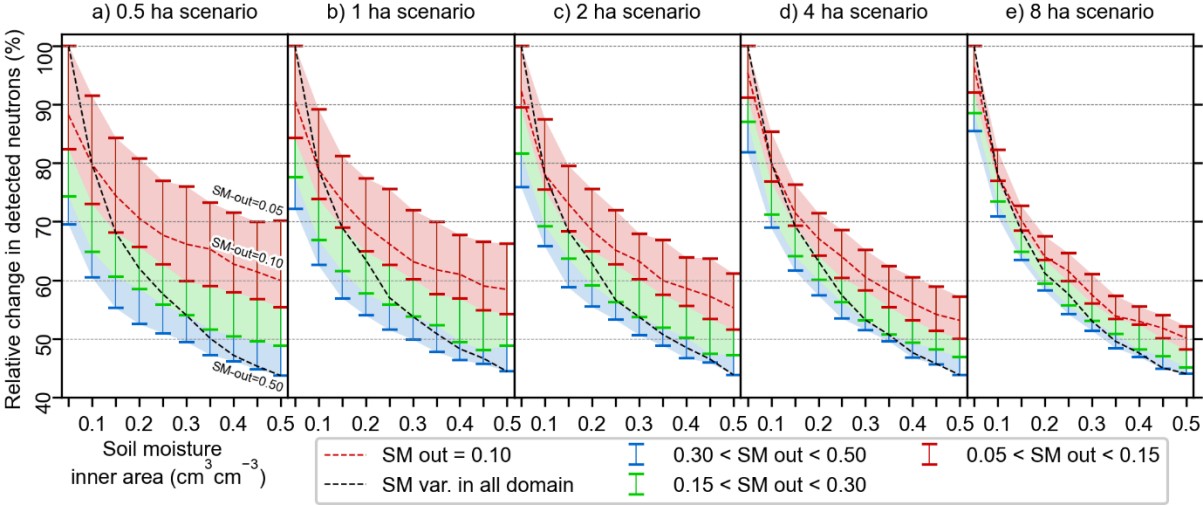

**Figure 9: Relative changes in detected neutrons due to SM variations in a) 0.5 ha, b) 1 ha, c) 2 ha, d) 4 ha, and e) 8 ha irrigated areas. The black dashed line shows SM variations that are homogeneous within the simulated domain. The red dashed lines and**
**the error bars indicate SM variations that occur only within the irrigated area for a different SM in the outer area.**

In the case of a 0.5 ha irrigated field, SM variations in the outer area generally induce a higher relative change in detected neutrons compared to SM variations that occur within the irrigated area. For example, a SM variation from 0.05 to 0.10 $cm^3cm^{-3}$ in the outer area (Figure 9a) induces a larger relative change (-11.8%) compared to the same SM variation in the





inner area (-8.6%). With larger irrigated areas, the influence of the outer area is reduced. In the 8 ha scenario, a SM variation

from 0.05 to 0.10 $cm^3cm^{-3}$ in the inner area induces a larger relative change in detected neutrons than that of a SM variation

from 0.05 to 0.50 $cm^3cm^{-3}$ in the outer area. These results suggest that it may be necessary to obtain SM information from

the surroundings of a target field in real-world applications of CRNS-based irrigation monitoring.

It is also of interest to reduce this analysis to a smaller range of SM in the outer area (i.e., 0.05 to 0.15 $cm^3 cm^{-3}$) as this is

generally the case in irrigated agricultural environments (shaded red area in Figure 9). In the case of 0.5 and 1 ha fields, a

SM variation of 0.10 $cm^3 cm^{-3}$ in the outer area (red error bars in Figure 9a-b) produces a relative change of the neutron

count rate that is higher than that of an irrigation event of 0.05 $cm^3 cm^{-3}$. This is not the case when the initial SM of the

domain is 0.05 $cm^3 cm^{-3}$ in the 2 ha scenario (Figure 9c), 0.05 to 0.15 $cm^3 cm^{-3}$ in the 4 ha scenario (Figure 9d), and 0.05 to

0.20 $cm^3 cm^{-3}$ in the 8 ha scenario (Figure 9e). Overall, even if the range of the SM in the outer area is reduced, it can still

have a considerable impact on the CRNS. This again underlines the importance of obtaining information on the SM in the

outer area, especially in case of relatively small irrigated fields.

## 4 Limitations and outlook

The results of this study suggest the feasibility of irrigation monitoring with CRNS for field dimensions and SM conditions

that are relevant in irrigation applications. However, it is important to emphasize here that the presented simulation results

constitute a best-case scenario as it assumes a square shape of the irrigated area, homogeneous SM changes in the inner

irrigated area, as well as stable SM conditions in the surroundings. Moreover, the possible influence of other environmental

conditions (e.g., vegetation, vertical SM distribution, heterogeneous land cover) is not considered. Real-world irrigation

monitoring may thus prove more challenging than what these results suggest as the accuracy of CRNS-based SM products

may suffer from heterogeneities in SM and in environmental conditions. For example, the role of within-field SM

heterogeneity should be further investigated as drip (Li et al., 2019), border strip, and lateral move irrigation can lead to an

uneven SM distribution which can be exacerbated by soil heterogeneity and surface roughness. Moreover, SM information

from the surroundings of a target field may be necessary to correct CRNS-based SM products not only in relatively small

fields (up to 2 ha) but also in larger ones. Real-world studies should thus assess in more detail the need for such information

in the correction of CRNS-based SM products for irrigation applications. For this, point-scale SM monitoring instruments

could be used to periodically retrieve SM values outside of a target field. Within this context, Monte Carlo simulations of

neutron transport represent an added value as they inform on the relative contribution from the surroundings of an irrigated

field to the neutron count. In addition, although simulations only offer a simplification of actual conditions, they could be

employed before installation to provide farmers with an estimate of the costs and benefits of a CRNS-based irrigation

monitoring system.



## 5 Conclusions

This study explores the feasibility of irrigation monitoring with CRNS by using Monte Carlo neutron transport simulations. Specifically, it investigates the influence of the moderator type, the dimensions of the irrigated area, and of the SM variations within and outside the irrigated area. Results show that a considerable fraction of the detected neutrons can originate from outside the irrigated region, which may represent a challenge in CRNS-based irrigation monitoring. This fraction varies considerably with the size of the irrigated area. Relatively higher contributions from the irrigated field are obtained for larger sizes of the irrigated area (e.g., 28% to 61% in 0.5 ha and 51% to 85% in 8 ha), higher SM outside the irrigated area, and, to a lesser degree, by detectors with thin HDPE moderators. The CRNS footprint depends both on the SM in the irrigated area and its surroundings as well as, to a lesser degree, on the moderator type and on the dimension of the irrigated area. Generally, detectors with thinner HDPE moderators result in smaller footprints and thus in stronger contributions from the irrigated field. However, thicker HDPE moderators and the addition of a thermal shielding result in higher relative changes in detected neutrons with respect to SM variations. Thus, such moderator types are expected to bring improvements in CRNS-based irrigation monitoring.

A CRNS with a 25 mm HDPE moderator and gadolinium shielding can detect irrigation events that increase SM by $0.05$ cm$^3$ cm$^{-3}$ even in fields as small as 0.5 ha when the SM in the entire simulated domain is $0.05$ cm$^3$ cm$^{-3}$. Detection is uncertain in a 0.5 ha field when initial SM was $0.15$ cm$^3$ cm$^{-3}$ or higher and again uncertain in 1 ha and 2 ha fields when initial SM was above $0.25$ cm$^3$ cm$^{-3}$. Higher detection chances are found in the case of irrigation events that increased SM by $0.10$ cm$^3$ cm$^{-3}$. Generally, larger irrigated fields, lower initial SM, and higher SM variations due to irrigation provided higher chances of detection. Using the same moderator, the relationship between relative changes in detected neutrons and SM variations for irrigated fields shows a reduced sensitivity compared to the case of homogeneous SM variations within the entire simulated domain. Such sensitivity reduction is due to the influence of SM from the outer area and is stronger in small irrigated fields compared to large ones. If SM in the outer area of 0.5 and 1 ha fields is between $0.05$ and $0.15$ cm$^3$cm$^{-3}$, it is generally not possible to distinguish whether a relative change in detected neutrons is due to irrigation or to SM variations in the surroundings. On the contrary, in an 8 ha field, irrigation-related SM variations of $0.05$ cm$^3$ cm$^{-3}$ can be identified up to a maximum SM of $0.20$ cm$^3$ cm$^{-3}$, independently from the SM of the surrounding area. The results suggest the importance of obtaining information on the SM of the outer area in real-world applications of CRNS-based irrigation monitoring.

Overall, this study shows that CRNS can be successfully employed in irrigation monitoring for both field dimensions and SM conditions that are relevant in irrigation applications. Nonetheless, real-world conditions may prove challenging due to the presence of additional factors that were not considered in this study. For example, heterogeneous SM variations within the irrigated field should be investigated in future studies. Moreover, the use of SM information from the surroundings of an actual irrigated field with limited size could be considered to correct CRNS-based irrigation products by combining actual





measurements with Monte Carlo simulations. Prior to installation, Monte Carlo simulations could also be employed to assess the costs and benefits of a given detector in a specific irrigated environment. In the long run, the combination of simulations

and real-world installations should be considered to establish CRNS as a decision support system for irrigation management and thus provide an additional tool to improve water use efficiency in agriculture.

**Appendix A**

In all the 500 simulations, the percentage of non-albedo detected neutrons varied with SM variations and with the type of moderator (Table A 1). Non-albedo neutrons represented a considerable percentage of the total detected neutrons. No

meaningful variations were obtained with different dimensions of the irrigated area. Thick moderators resulted in higher percentages of detected non-albedo neutrons compared to thin moderators (e.g., 5.5 to 9.9 % with a 5 mm HDPE moderator and 15.2 to 31.8 % with a 35 mm HDPE moderator).

**Table A 1: Minimum and maximum percentage of non-albedo neutrons detected with different moderators and different scenarios. The colour in each cell indicates the mean percentage with low values in white and high values in grey.**

| Moderator | Scenario (inner area dimension) | | | | |
|---|---|---|---|---|---|
| | 0.5 ha | 1 ha | 2 ha | 4 ha | 8 ha |
| 5 mm | 5.5 - 9.9 | 5.5 - 10.1 | 5.5 - 9.8 | 5.6 - 9.7 | 5.8 - 9.9 |
| 10 mm | 7.4 - 14.5 | 7.4 - 14.7 | 7.3 - 14.5 | 7.5 - 14.4 | 7.8 - 14.6 |
| 15 mm | 9.1 - 18.5 | 9.1 - 18.6 | 8.9 - 18.7 | 9.2 - 18.5 | 9.4 - 18.7 |
| 20 mm | 11.1 – 23 | 11.1 - 22.9 | 10.9 - 23.3 | 11.2 - 23.0 | 11.3 - 23.2 |
| 25 mm | 12.2 - 25.6 | 12.2 - 25.5 | 12.1 - 25.8 | 12.4 - 25.5 | 12.4 - 25.8 |
| 25 mm + Gad. | 12.1 - 27.2 | 12.1 - 27.1 | 11.9 - 27.5 | 12.3 - 27.2 | 12.3 - 27.3 |
| 30 mm | 13.7 - 28.7 | 13.7 - 28.6 | 13.6 - 29.0 | 13.9 - 28.5 | 13.9 - 29.0 |
| 35 mm | 15.2 - 31.5 | 15.2 - 31.3 | 15.0 - 31.8 | 15.4 - 31.3 | 15.3 - 31.8 |

**Appendix B**

Thermal neutrons were found to have a smaller footprint compared to epithermal neutrons (Jakobi et al., 2021). The number of detected thermal neutrons and their percentage among all detected neutrons changes abruptly when different moderators are used (Figure B 1). A moderator of 5 mm HDPE results in thermal neutrons being above the 50% level of detected neutrons. Such percentage can be reduced by increasing the thickness of the HDPE moderator and by adding a gadolinium

shielding. Moderators that detect more thermal neutrons will result in smaller footprint compared to other detectors (see Figure 4 and Figure 5) and, likely, in more detected neutrons originating within an irrigated field (see Table 2).





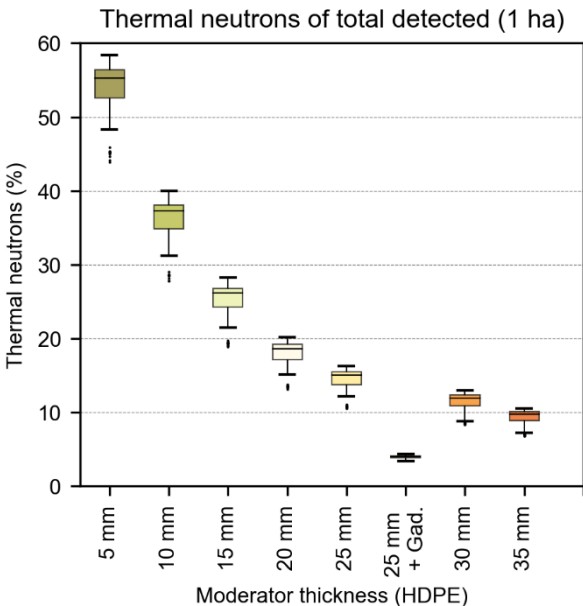

**Figure B 1: Boxplot of the percentage of detected thermal neutrons over total detected neutrons with different moderator types for the 1 ha scenario (each boxplot shows the results of 100 simulations).**


Table B 1 shows that the variation in detected thermal neutrons that originate within a 1 ha irrigated field is rather small with a 5 mm HDPE moderator (although it should be noted that the relative errors are not negligible as detected thermal neutrons range between ~700 and ~800). Consequently, thermal neutrons can concentrate the instrument footprint over a small area (e.g., an irrigated field) but this might not be followed by an added value in the detection of SM changes within such area.


**Table B 1: Thermal neutrons with origin in IN_1ha detected with a 5mm HDPE moderator and expressed as a fraction of the maximum value (i.e., 840 thermal neutrons detected with SM of 10 $cm^3$ $cm^{-3}$ in IN_0.5ha and 0.05 $cm^3$ $cm^{-3}$ in OUT_0.5ha).**

|  |  | SM ($cm^3$ $cm^{-3}$) in the 1 ha field | | | | | | | | | |
|---|---|---|---|---|---|---|---|---|---|---|---|
|  |  | **0.05** | **0.10** | **0.15** | **0.20** | **0.25** | **0.30** | **0.30** | **0.40** | **0.45** | **0.50** |
| | **0.50** | 0.92 | 0.95 | 0.98 | 0.90 | 0.92 | 0.91 | 0.89 | 0.81 | 0.85 | 0.84 |
| | **0.45** | 0.94 | 0.96 | 0.96 | 0.95 | 0.97 | 0.91 | 0.89 | 0.86 | 0.84 | 0.79 |
| | **0.40** | 0.91 | 0.96 | 0.99 | 0.96 | 0.94 | 0.92 | 0.88 | 0.86 | 0.82 | 0.84 |
| **SM in outer area** | **0.35** | 0.94 | 0.93 | 0.98 | 0.98 | 0.94 | 0.93 | 0.86 | 0.85 | 0.86 | 0.83 |
| | **0.30** | 0.92 | 0.95 | 0.98 | 0.95 | 0.97 | 0.92 | 0.87 | 0.86 | 0.81 | 0.82 |
| | **0.25** | 0.91 | 0.98 | 0.96 | 0.97 | 0.97 | 0.89 | 0.91 | 0.87 | 0.82 | 0.82 |
| | **0.20** | 0.91 | 0.95 | 0.95 | 0.94 | 0.95 | 0.90 | 0.88 | 0.87 | 0.83 | 0.83 |
| | **0.15** | 0.90 | 0.96 | 0.99 | 0.97 | 0.96 | 0.94 | 0.92 | 0.89 | 0.83 | 0.81 |
| | **0.10** | 0.94 | 0.95 | 0.97 | 0.95 | 0.94 | 0.89 | 0.89 | 0.84 | 0.83 | 0.81 |
| | **0.05** | 0.95 | 1.00 | 0.98 | 0.96 | 0.92 | 0.92 | 0.88 | 0.84 | 0.85 | 0.82 |



**Data availability**

Data are available upon contacting the authors. The URANOS model can be downloaded at the following address:
http://www.ufz.de/uranos (last access: 29.06.2022).

**Author contributions**

Conceptualization CB HRB MK, methodology CB HRB MK, investigation CB MK, methodology finalization CB HB MK JAH HJHF OD, writing - original draft preparation CB. writing - review and editing CB HB MK JAH HJHF OD. All authors have read and agreed to the published version of the manuscript.

**Competing interests**

M. Köhli holds a CEO position at StyX Neutronica GmbH.

**Acknowledgements**

This research received the support from the ATLAS project funded through the EU's Horizon 2020 research and innovation program under grant agreement No. 857125 and from the DFG 425 (German Research Foundation) via the project
357874777, research unit FOR 2694 Cosmic Sense.

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
