# Peer review of "Feasibility of Irrigation Monitoring with Cosmic-ray Neutron Sensors"

_Geoscientific Instrumentation, Methods and Data Systems, 2022_

## Referee Comment (RC2)

This paper investigates the feasibility of CRNS-based SM monitoring in irrigated environments. The paper is informative with lots of simulations in different scenarios. Simulation of neutron count in different scenario was performed with Monte Carlo simulations and Ultra Rapid Adaptable Neutron-Only Simulation. However, obtained results were not validated using real case scenario of irrigated areas.

The author chose a square, not a circle, irrigated area. While the CRNS is tube-shaped and the footprint of the CRNS is circular, is there a reason for that specific shape? Also, do we expect an improved detection rate or minimal if he changed it to circular?

Belo  few questions that could make the paper clearer for the reader:

Line 14:  the unit needs typo correction

Introduction: Recent work on soil moisture mapping from SAR images is worth reporting in the introduction, primarily those that provide operational soil moisture mapping through the synergistic use of Sentinel-1 and Sentinel-2.

In line 143-144, you mentioned that variations in humidity, vegetation, and other environmental variables can affect the footprint but with less degree than the SM effect. However, in the simulation, these factors were fixed (line 167-170). Can you explain what the impact would be on the simulation results if you include the diurnal weather conditions?

Line 187:  why did you choose 9m radius specifically for all the simulations? Is there a method for choosing the right radius size for the tube?

Looking at Table2 + Figure 3 and then Figure 6, Figure 7 and Fig 9. They are very well connected, I wonder why they did not come directly after each other so that it is easier for the reader to stay on track!

Lines 374 – 376: *"As shown in Figure 8, an irrigation event that leads to a 0.05 cm3cm-3 increase in SM can be detected with CRNS (relative change in detected neutrons higher than 3σ + α) when the initial SM of the simulation domain is 0.05 cm3cm-3."*  It looks like there is an overlap in this figure: the green bars represents the 0.05 cm3cm-3 initial SM and the blue bar for the 0.10 $cm^3cm^{-3}$. Indeed, the relative change is lower when the increase of soil moisture is higher. Is that correct?

---

## Author Comment (AC1)

**Response to reviewers**

**Anonymous Reviewer n.1**

This study investigates the feasibility of soil moisture (sm) detection with cosmic-ray neutron sensors (CRNS) in the center of a squared irrigated field. The authors investigate the sm-neutron relationship and the footprint radius for various scenarios of detector shielding material, field sizes, and combinations of soil moisture conditions in the inner and the outer areas. They conclude that different thickness of HDPE material influences the footprint radius and the detector sensitivity, with HDPE 25mm + Gadolinium leading to the highest sensitivity to soil moisture changes and to the largest footprint radius. Moreover, they found that soil moisture from the outer fields substantially influences the CRNS sensitivity to the irrigated field, especially for small fields.

The study is very well written and highly relevant, being the first of its kind to study various detector thicknesses and gadolinium shields with regards to the neutron-soil moisture relationship. This will be a very important information for CRNS users and suppliers in order to significantly improve the sensor performance. Furthermore, the investigation of irrigation support is also very relevant for current and future CRNS applications. However, this study is only theoretical and limited to a very specific geometry (CRNS within a squared irrigated field). Nevertheless, it is a good step forward.

Major concerns with this study should be clarified before publication of this manuscript, particularly regarding the model assumptions, the consistency of the argumentation, and the integrity and applicability of the conclusions. Please address also the minor comments and the specific comments below.

*We sincerely thank the reviewer for the appreciation of our work and for the time spent to produce a thorough and useful review. We have carefully considered all comments and observations. We provide here replies to each point with indications on the changes that were included in the new version of the manuscript or motivations for not applying further edits when this was not deemed necessary.*

*We hope that our work can address all the points raised by the reviewer and that this new version of the manuscript can be accepted for publication in Geoscientific Instrumentation, Methods and Data Systems.*

**Major concerns:**
1. The authors seem to change the water content of the irrigated soil in the whole soil column (up to 160 cm depth). However, it is highly unrealistic to assume vertically homogenous wetting of the soil due to surface irrigation! Hence, I wonder how useful the results of this study are for practical applications. In the real world, surface irrigation adds moisture only to the first few centimeters of the soil, or decimeters, depending on typical depth of the roots and the hoses. This would leave a large part of the vertical footprint dry, such that the results obtained in this study might significanlty overestimate the actual sensitivity of CRNS to irrigated soil.

*We thank the reviewer for this observation since, although known to the authors, this limitation was only briefly mentioned in section "4 Limitations and outlook". We agree that the wetting due to precipitation or irrigation is not homogeneous with depth. The wetting caused by irrigation would most probably start at the soil surface and reach the deeper soils within a certain amount of time depending on several variables (see below). Typically, we can expect deeper soils to have a higher SM compared to the soil surface when irrigation starts. Then, SM would increase in the soil column and peak at the top of the profile. Which would be followed by a redistribution of SM that can take up to several days/weeks.*

*We assumed a homogeneous SM distribution for our study because we considered that SM dynamics in soils are a rather complicated subject. The SM distribution in the soil column before, during, and after an irrigation event can depend on (among other factors):*
  *- Soil texture, porosity, bulk density, and presence of biopores.*
  *- Previous SM conditions within the soil profile and depth of the water table.*
  *- Root depth and distribution, vegetation type, and root water uptake.*
  *- Agricultural management (ploughing depth and irrigation type).*
*Thus, to define how the SM varies in depth in an agricultural area, it would be necessary to consider at least soil depth and texture, water table depth, ploughing depth, rooting depth, and root distribution.*

*Another aspect to consider is the variation of the depth of investigation of a CRNS with SM variations as well as with increasing distance from the instrument (larger below the instrument and relatively shallower at distance). Thus, the overestimation of the contribution of irrigation to the signal would be present mostly in the vicinity of the instrument. Also, during an irrigation event, the wetting of the upper soil would reduce the depth of investigation of the CRNS and thus diminish the relative contribution of deeper soils to the signal. We believe that these two factors could reduce, although not completely remove, the overestimation effect mentioned by the reviewer.*

*Overall, the study of the influence of vertical heterogeneity would add a further dimension to the problem that is investigated in this study and would significantly increase its complexity. As it is not feasible to include all possible scenarios into the simulation of a CRNS framework, this theoretical study presents one particular solution to one subset of problems, which we believe is already of great interest. Further research is necessary (and currently carried out by some research groups) to integrate vertical heterogeneity based, for example, on hydrological models. But this again is, in our opinion, outside the scope of this study.*

*This said, the observations of the reviewer are valid, and we agree that we did not discuss this specific limitation in a sufficient way in the previous version of the manuscript. In the new version of the manuscript, we will point at the limitation of a homogeneous SM distribution and at the possible overestimation effect in section "4 Limitations and outlook"*

2. The authors conclude that 25 mm HDPE+Gd is the best shielding variant for CRNS detectors (L359) without studying other HDPE thicknesses together with Gd. It is indeed interesting to see gadolinium used for the first time in a sophisticated analysis of detector sensitivity to footprint and soil moisture changes. However, the authors used Gd only in

the 25 mm HDPE setup. In order to distinguish and better understand the effects of HDPE and Gd on the variables of interest, and to find the best performing detector design, it would be necessary to simulate Gd shielding also for different HDPE thickness.

*We agree with the reviewer that investigating the effects of a Gadolinium based shielding on different moderator thicknesses would be interesting, at least in general terms. However, to our knowledge, such shielding is currently commercialized only with 25 mm HDPE moderators. Additionally, as shown in the following* Figure 1, *a Gadolinium based shielding prevents the detection of approximately 90% of thermal neutrons. This effect differs from a HDPE moderator since the sensitivity is not shifted towards higher energies but rather cut in the thermal range. Thus, it can be expected that a 25 mm HDPE moderator with Gadolinium shielding would generally detect less neutrons than a 25 mm HDPE moderator without Gadolinium shielding.*

[Figure]

*Figure 1: Response functions with the thermal neutron energy range marked with a red area.*

*We could then expect that, by adding Gadolinium shielding to a thin moderator that has high sensitivity to thermal neutrons and low sensitivity to epithermal and fast neutrons (e.g., 5 mm HDPE), the number of detected neutrons would decrease considerably (partial removal of the red shaded area in* Figure 1*). In thicker moderators, the work of Weimar et al. (2020) shows that for moderator thicknesses of 20, 22.5, 25, and 27.5 mm HDPE there are only small differences when the Gadolinium shielding is added. The authors concluded that a 22.5 m HDPE moderator with a Gadolinium thermal shielding provided the best overall performance but with only a marginal difference compared to a 25 mm HDPE moderator (as proposed by Desilets et al. (2010)) with Gadolinium shielding. It is true that thinner moderators yield slightly better signal dynamics at the cost of a small reduction of the count rate. However, such differences are marginal and, in our opinion, not significant when put in the framework on a study such as the one proposed here.*

*In the end, to our knowledge, a 25 mm HDPE moderator with Gadolinium shielding is the common choice because of a) recommendations from, for example, Desilets et al. (2010) confirmed by Weimar et al. (2020) and b) production costs. Previous research has shown the influence of different moderator thicknesses, at least in a range of moderators that is relevant. We thus believe that further investigations on this topic would go out of the scope of the study and not add novel information for the reader.*

*Nonetheless, we carefully considered the comment of the reviewer, and we will elaborate on the choice of a 25 mm HDPE moderator with Gadolinium shielding in the material and methods section. In addition, we will better specify in the conclusion of the manuscript that Gadolinium shield variant was only studied for a 25 mm HDPE moderator as this was not often specified in the previous version of the manuscript.*

3. The authors conclude that outer sm is an important factor for irrigation monitoring with CRNS by considering only one shielding option that provides the largest footprint. In section 3.4 and 3.5 the impact for inner sm changes on the neutron detector sensitivity is investigated, but only for 25 mm HDPE-Gd (mentioned in 3.4, not mentioned in 3.5). In the same study, the authors find that different shielding options have different footprint radii. So, I would expect that shielding options with lower footprint radii would lead to less impact of the outer sm. Please elaborate on this and whether different shielding options could be recommended for smaller fields than for larger fields to reduce the influence. This might change your conclusion. Also, use this opportunity to join the two otherwise separate parts of "shielding/footprint analysis" and "sm/area analysis", in order to demonstrate that a combination of those two studies in a single paper could actually lead to a greater benefit.

*In the new version of the manuscript, it will be clearly stated in section 3.5 that the analysis is performed for a 25 mm HDPE moderator with Gadolinium Shielding. This was not the case in the previous version of the manuscript, and we thank the reviewer for highlighting it.*

*It is true that the Gadolinium shielding provided a footprint that is larger than that of the other moderators. However, it should be noted that the R86 of the 25 mm HDPE moderator with Gadolinium shielding was approximately 5 to 15 m larger than that of the other moderators (with R86 values between ~120 and ~270 m, see Fig.4 of the manuscript). Here, we noted that we did not clearly state in the previous version of the manuscript that such difference in R86 dimension is rather small and we will mention it in the new version.*

*Additionally, in Fig.6 of the manuscript, we show how the 25 mm HDPE with Gadolinium shielding offers a higher sensitivity compared to the other moderators. Thus, although a different moderator with a smaller R86 (e.g., 5 mm HDPE) could possibly lead to a reduced influence of the SM outside the investigated field, the moderator with the highest sensitivity (25 mm HDPE with Gadolinium shielding) would still be preferrable. Therefore, we decided to focus the second part of the results on such moderator only. In sections 3.4 and 3.5, we could have included results from all eight moderators, and we initially tried to pursue such path, but we soon realized that this resulted in an overwhelming and confusing amount of information with no real benefit for the overall message of the study.*

*We nonetheless think that the reviewer is raising a fair point and decided to provide additional figures in our reply to the comments. The following* Figure 2 *was produced starting from Fig.9 of the original manuscript (Part-i). To this, we added similar figures obtained with data from a 25 mm HDPE moderator (Part-ii) and 5 mm HDPE moderator (Part-iii).*

**Part-i - 25 mm HDPE + Gadolinium**

[Figure]

**Part-ii - 25 mm HDPE**

[Figure]

**Part-iii - 5 mm HDPE**

[Figure]

*Figure 2: Part-i is a copy of Fig.9 of the manuscript (referred to a 25 mm HDPE moderator with Gadolinium shielding. Part-ii and Part-iii are figures similar to Fig.9 of the manuscript built with data from a 25 mm HDPE and 5 mm HDPE moderators respectively.*

*As visible in* Figure 2, *the use of a 25 mm HDPE moderator without Gadolinium shielding, which has an R86 that is 5-10 m smaller than the version with Gadolinium, does not show apparent reductions of the influence of the SM outside the irrigated field. However, there is a general reduction in sensitivity, as can also be seen in Fig. 6 of the manuscript. Furthermore, a 5 mm HDPE moderator, although having a footprint that is sometimes 20 m smaller than that of the 25 mm HDPE with Gadolinium variant, does not show a reduction of the effect of the outside SM. Such effect of the outside SM sometimes appears higher than in the Gadolinium case, but the most important difference is the generally lower sensitivity of the moderator without gadolinium.*

*We believe that this could be connected to the fact that the detected neutron intensity of a CRNS peaks nearby the instrument and decreases with distance. Thus, a small variation in the R86 alone might not lead to meaningful differences between moderators and the instrument with higher sensitivity to SM changes should be preferred. We now realize that this concept was not fully described throughout the manuscript. It will be however better introduced in the new version of the manuscript. Also, we think that the results of* Figure 2 *would be a meaningful addition to the manuscript and we will produce a new appendix to show such results.*

4. The authors suggest that sm measurements outside the irrigated area are needed to properly estimate sm inside with CRNS. But in this case, what is the added value of CRNS compared to using the suggested number of additional sensors and putting them inside from the start?

*We carefully considered the comment of the reviewer and we realized that we did not offer enough information in the previous version of the manuscript. What we could suggest is the installation of a SM sensor outside the irrigated field or, alternatively, the use of a portable device. This scenario of CRNS plus a single device is different compared to the use of a sensor network within the irrigated field as a) only the CRNS would be installed in the field, b) portable or point-scale SM measurement devices are relatively inexpensive, and c) measurements of the SM outside the irrigated field could provide correction for multiple CRNS installed in fields that are located within a large area. Following the comment of the reviewer, the new version of the manuscript will mention such considerations in section 3.5, in section 4, and in the conclusions.*

5. The study is limited to one specific case of field geometry: a CRNS detector in the center of a square-shaped irrigated field surrounded by a homogeneous field on all sides. Although simulations of various shielding types and square sizes are scientifically interesting, the transferability of the results to practical field geometries might be very limited. I would recommend to better communicate this limitation, at first by improving Figure 1 with a detector symbol in the center and scales in meters. Second, by providing real-world examples where this scenario could be applicable (I would rather think that radial field geometries would have been a better choice, e.g. for pivot irrigation). Third, by discussing potential deviations and uncertainties of the results if the sensor location would not be ideally centric, or if the field shape is a circle instead of a square. This would allow users to get a better idea whether the results would be still applicable to a certain degree, or whether completely new simulations would be necessary for every single deviation from the presented ideal case.

*We will include the suggested changes in Fig.1 (see following* Figure 3*) and, as described in detail below, we will add considerations on the selected shape of the irrigated field in section 2.4 "Simulations setup" and in section 4 "Limitations and outlook".*

[Figure]

Figure 3: alternative figure that substitutes Fig.1 in the new version of the manuscript.

*Reviewer n.2 also points at the shape of the irrigated field. We believe that a circular pivot irrigation field would have not been the best choice for this study. This because circular centre-pivot irrigated fields are very large and typically 400 m in radius (although 500 m are also common and larger exist). Small centre-pivot irrigated fields are not common. The large dimension of a centre-pivot irrigated field means that most of the neutrons detected by a CRNS placed in the middle of such large field (if not all detected neutrons) originate within the irrigated field. Thus, the irrigation that the CRNS would sense is comparable to a rain event, and we think this would not be as interesting as a smaller field of a few ha where the outside area plays an actual role.*

*Relatively small fields as those investigated in this study are most often rectangular and not circular. It is true that the rectangular shape can vary greatly and elongated rectangular shapes are common, but the inclusion of elongated shapes in this study would lead to a much more complex manuscript and would go beyond the scope of the manuscript. Since the manuscript is already long and complex in the current form, we believe that the reader would have strong difficulties in navigating through additional shapes of the irrigated area. Nonetheless, it could be expected that such elongated shapes would be more challenging for the CRNS compared to a*

*squared shape. We agree with the reviewer that the selection of the shape is an important topic, and we will include additional considerations in section 4 "Limitations and outlook". We also think that future research, especially in real-case scenarios, should investigate these aspects and we will mention this in the revised manuscript.*

*Furthermore, we tested the simulation of a circular and rectangular (142x70 m) irrigated fields of 1 ha area and compare the results with those of a squared 1 ha irrigated field. The results are shown in* Figure 4 *in a similar way as they are shown in Fig.8 of the manuscript. In general, differences between the squared, circular, and rectangular shaper are rather small, at least in this simulation setup. Compared to the squared shape, there might be sometimes a small tendency towards higher relative changes in detected neutrons for a circular shape and a tendency towards lower relative changes when a rectangular shape is used. However, results are too similar to draw meaningful conclusions. A clearer picture could possibly be obtained if these two additional scenarios are simulated for the entire soil moisture range of 0.05 to 0.50 cm3 cm-3 and for the five investigated areas of the irrigated field. However, this would increase the number of simulations by +200%, which is not feasible due to time and computational constraints and goes beyond the scope of the manuscript. Thus, we decided not to add simulations of different field shapes to the manuscript.*

[Figure]

*Figure 4: CRNS chance of detecting irrigation events of 0.05 and 0.10 cm3 cm-3 (blue and green bars respectively) in a) squared, b) circular, and c) rectangular irrigated field of 1 ha. The bars show the relative change in detected neutrons induced by the irrigation event while the dashed lines show the prescribed detection thresholds. The red area below the σ+α threshold indicates uncertain detection.*

Regarding the positioning of the CRNS, we think that the assumption that the instrument is positioned in the centre of an irrigated field is a common practice. CRNS are sometimes not positioned in the centre of a field, for example when multiple CRNS are placed in different areas of a single field. But we believe that the discussion and investigation of device placement that are not in the centre of a given field (or its vicinities) as well as the placement of multiple devices goes beyond the scope of this study. Regarding the need for new simulations prior to CRNS installation, we suggest in the original version of the manuscript (section 4 "Limitations and outlook" and conclusions) that neutron transport simulations could be useful to assess costs and benefits before CRNS installation as well as to quantify the partitioning of neutron origins in real case scenarios. We thus do not believe that these simulations are currently exhaustive, and we think that we did not suggest their exhaustivity in the manuscript. We will however strengthen the discussion of these

*limitations in section "4 Limitations and Outlook" and point at possible future research and studies in this regard.*

**Minor comments**
1. The authors mention that sm change outside can be larger than sm change from irrigation inside. However, it is not clear in which scenarios it is actually realistic to assume that sm changes "only" outside. Most of the time, precipitation occurs within CRNS footprints rather homogeneously. So the only cases I could image of sm changing outside (and not inside) is that these fields are also irrigated, and with a completely different schedule. You could communicate more openly that this is a special case, and only in this case the sm-outside issue becomes relevant.

*We agree with the reviewer, and we understand now that we created some degree of confusion in certain instances. The new version of the manuscript clarifies this aspect at the beginning of paragraph 3.5 where we now mention the irrigation of only the neighbouring fields.*

2. The authors seem to assume that only neutrons that originated from the inner area carry its sm signal (L211). However, Köhli et al. 2015 mentioned a few intermediate interactions of neutrons from longer distances with the soil on their way to the detector. In this context, it seems that neutrons from outer regions could carry signals from the inner regions, too. In this case the authors are encouraged to reconsider their assumption.

*We now understand that we presented this assumption in a rather simplified way. In the new version of the manuscript, we clarify that the neutrons that originate within the irrigated area carry a large portion of the information of interest. In our opinion, this does not change the general message that a large percentage of detected neutrons that originate outside the irrigated field is a challenge in irrigation monitoring with CRNS.*

*In the new version of the manuscript, we will extend section 2.6 which will read "Here, it was assumed that the neutrons that originate within the inner irrigated field carry the bulk of the information of interest. Although neutrons that originate outside the irrigated field can have occasional within-field interactions before reaching the CRNS (Köhli et al. 2015), these were considered of secondary importance for the scope of this study". Similarly, section 3.1 will read: "This percentage represents the detected neutrons that originate within an irrigated field and thus carry the bulk of the information of interest in case of irrigation applications".*

3. Minimum soil moisture used in this study is 0.05 m³/m³, while irrigation might be particularly interesting in extremely arid regions, where sm below 0.05 can exist. Given the very steep neutrons-sm function, there is a potentially significant performance increase of CRNS for dry soils. Can you add ~0.03 m³/m³ to your analysis?

*We carefully considered the possibility to add 0.03 m³/m³ to our simulations. But we finally concluded that this would not add important information or change the message and conclusions of the manuscript. Although it is true that soils can theoretically have a SM content below 0.05 m³/m³ (e.g., residual soil moisture), we believe this value is very unlikely in agricultural fields (if not impossible in irrigated fields). Regarding the area that surrounds an irrigated field, we again*

*believe that a SM of 0.05 m³/m³ is already a rather low value. A lower value could be found in the top 1-2 cm of soil, but then higher SM would be commonly found below such shallow depth. Thus, we believe that a homogeneous value of 0.05% in the soil column is already a sufficiently low SM value for the scope of this study.*

*Regarding the results that we could expect by including a SM of 0.03 $cm^3$ $cm^{-3}$, (see, for example, Fig.8 of the manuscript) an irrigation event that leads to a final SM of 0.08 $cm^3$ $cm^{-3}$ (plus 0.05 $cm^3$ $cm^{-3}$ as done for the other SM cases) would be easily sensed by the CRNS since the SM variation 0.05 to 0.10 $cm^3$ $cm^{-3}$ is already sensed. Same can be said for a 0.10 $cm^3$ $cm^{-3}$ SM variation (from 0.03 to 0.13 $cm^3$ $cm^{-3}$). At the same time, we do not think that it is necessary to investigate a SM increase from 0.03 $cm^3$ $cm^{-3}$ to 0.05 $cm^3$ $cm^{-3}$ as this does not seem relevant in irrigation.*

4. Nomograms for the presentation of the results are very hard to read (e.g. Fig 3). I can understand the authors idea to put both, the inner and the outer sm on the two axes, but it took me several minutes starring at the plot to understand what they are showing. And now that I understand it, I still find it hard to read out what fraction of neutrons comes from the inner or the outer part for a given soil moisture condition. Especially since both relative neutrons are not adding up to 100% (due to direct neutrons?). So, I would strongly suggest to reconsider these graphs, focusing on the main message, which probably is: "How do neutrons from inner and outer areas compare?". If two variables need to be compared, try to show them in the same plot. And since they add up to a total neutron count (or to 100% inlcuding direct neutrons), the usage of stacked barplots might be good choice. One advantage of a N-over-SM plot would also be that the curves could be easily compared with the conventional N(SM) functions to show how this function changes for different irrigation pattern. Just like in Figure 9. Consider replacing Fig 3 with Fig 9 or at least refering to it.

*We understand now that the readability of Fig.3 can be improved. We also agree with the reviewer that clear indication on the presence of non-albedo neutrons in the calculation should have been included in the description of the figure.*

*For the new version of the manuscript, we first attempted to create a figure based on stacked bar plots as this was one of the possible suggestions from the reviewer. The result can be seen in the following* Figure 5*. Although some details of the figure could be improved, we think that this illustration method does not allow to clearly show many simulation results. Although the difference between the 0.5 and the 8 ha cases is crystal clear (left and right column respectively), the differences in neutron origin between the 100 simulations for a 0.5 ha field is hardly readable. Same goes for the 8 ha simulations.*

[Figure]

*Figure 5: Alternative to Fig03 of the original manuscript using stacked bar plots. Not selected.*

*Nonetheless, we still agree with the reviewer comment on Fig.3 and tried an alternative illustration for Fig.3 of the manuscript. We followed the suggestion of using Fig.9 style and the result is the following* Figure 6.

[Figure]

Figure 6: alternative figure that substitutes Fig.3 in the new version of the manuscript.

*In this figure, we decided to not show the results of all simulations as the plotted lines would sometimes get too close to each other. Thus, we only show results for three values of SM outside the irrigated field, which we believe is sufficient. We also show only detected albedo neutrons now, and thus the percentages sum is 100%. We think that this new figure is more easily readable, and it conveys the messages that are described in the text. We thank the reviewer for his suggestions. Following this change, the text and the figure caption will be modified to fit the new Fig.3.*

5. Parts of the introduction are unnecessary or not clear. There are long paragraphs about food security and irrigation in general, point-to-large scale sensors, CRNS detectors in general and their multifaceted applications from snow to regional modeling, and so forth. All this sounds like a great literature review, but it is out of scope in large parts. Instead, in the end of the introduction the very important concept of "energy dependent response function" is just mentioned, while it has never been introduced. Please consider shorteing the introduction and provide a more concise structure with a focus on the actual topic: simulation of multiple neutron detector variants in heterogeneous irrigated terrain. Unclear: the argumentation about thermal neutrons, how are they different from epithermal neutrons, and why is it necessary to exclude them? Here would be a good spot to elaborate on energy response functions. See also the 16 specific comments below.

*We will shorten the first two paragraphs of the introduction to have a better focus on the topic of the manuscript. Also, we will better introduce and describe certain topics such as the difference*

*between thermal and epithermal neutron detection. The specific comments provided by the Reviewer will be also addressed as it is described in the followings.*

**Specific comments:**
- L47: I don't see how Andreasen et al. 2016 demonstrates that CRNS can close gap between point and large scales. Please double-check the reference and think about providing references related to the footprint and in comparison with actual point measurements (for example, Heistermann et al. 2021, doi:10.5194/hess-25-4807-2021)

*The original reference will be substituted with the one proposed by the Reviewer.*

- L49: "cosmic-ray radiation" is tautologous.

*In this case, "cosmic-ray radiation" is not tautologous as it refers to radiation induced by cosmic radiation. Cosmic rays are (primarily) high-energy protons. By interaction with the atmosphere, they create showers of secondary particles which are, in case of neutrons, a different type of radiation. Therefore, the term "cosmic-ray induced neutrons" is often used instead of the shorthand versions "cosmic-ray neutrons" or the mentioned "cosmic-ray radiation". We therefore believe that no modification needs to be made in this point of the manuscript.*

- L50: "The detected neutrons are generally in the thermal ... or epithermal ...". Please rephrase to avoid questions like: What means "generally"? Can a detector directly detect epithermal neutrons? Is a spectrometer involved? Can you provide a reference for the energy sensitivity?

*In the introduction, we provide a simplification of the actual processes that are involved in neutron detection, both technically and physically. In Weimar et al. (2020) the principles of neutron detection for a CRNS instrument are described at length. After considering the comment of the reviewer, we decided to add a reference to Weimar et al. (2020) in this sentence. Specifically, the largest part of detected neutrons lies within the given energy ranges. More importantly, most of the details are included in the simulation and therefore do not alter the results. Other details which are not represented in the simulations are, for example, the actual geometry of the sensor. Francke et al. (2022) show however, that the differences between a simulated virtual detector and a simulated detector with its actual dimensions and materials are rather small. We believe that the text is sufficiently informative here after the addition of the abovementioned reference.*

- L52: Consider adding a references that is less than 10 years old, e.g., Köhli et al. 2021, doi:10.3389/frwa.2020.544847

*The reference will be added to the text.*

- L56: "dry soils have a higher neutron density" - Please rephrase. Is it the soil that "has" this neuton density, or the atomic nuclei inside the silicate atoms? Or the air above the soil?

*We will modify the text in: "For example, the count rate is inversely proportional to SM since dry soils result in higher environmental neutron density that allows more accurate measurements compared to wet soils".*

- L49-57: The whole paragraph: you start with telling that CRNS is both, thermal and epithermal. Are all these statemens in the text (sensitivity to sm, snow, vegetation, ...) refererring to thermal, to epithermal, or both? Do they behave equally? If no, how are thermal neutrons different? If yes, why do you want to exclude them?

*Following the comment of the reviewer, we will describe at the end of the paragraph that, in the specific case of soil moisture monitoring, thermal neutrons have been shown to have lower sensitivity to soil moisture than epithermal neutrons and that thermal contamination of standard probes leads to a lower signal-to-noise ratio compared to shielded detectors. We will also provide references to such statements. Such addition will be limited to soil moisture as this is the focus of our study. We believe that a wider description would further lengthen the introduction and go beyond the scope of this study.*

- L60: I don't see how Schrön et al. 2017 uses multiple counter tubes to achieve higher count rates. Do you mean Schrön et al. 2018, doi:10.5194/gi-7-83-2018 ?

*We will substitute the original reference with the proposed one.*

- L64: "... prevent the detection of thermal neutrons" - Not clear: why should the detection of thermal neutrons be prevented? Please restructure this paragraph to explain why thermal neutrons behave differently and why they are not useful for soil moisture monitoring.

*For the answer to this comment, please refer to the above comment L47-57.*

- L69-70: You argue that CRNS does not need to be removed during harvest. Is the instrument of infinitesimal size? If not, what is the spatial extent of the apparature including anchoring, and how can a farmer pull around it? Please consider reporting about pros and cons of CRNS more objectively.

*The previous version of the manuscript described this part in a rather simplistic way, which may be interpreted as an underestimation of CRNS cons. Indeed, the instrument is not infinitesimal and occupies a space of approximately 35x35 to 80x80 cm, depending on the instrument type, manufacturer, and installation. Our goal was to highlight how a single CRNS differs from a sensor network composed of multiple distributed nodes. In this case, the farmer needs to drive around a single sensor (generally well visible and placed in the middle of the field) instead of driving around multiple sensors that are distributed within the field. In the latter case, the complication of driving around multiple sensors generally results in the need for complete removal and reinstallation. This is not the case with a single CRNS. We will improve the text in the new version of the manuscript by including additional wording from Franz et al. (2016): "In the context of agricultural applications, a CRNS can be placed in between or out of the way of routine production practices. It consequently does not present the logistic challenges associated with directly inserted sensors,*

*which need to be removed and reinstalled during harvest, planting, and other management actions (Franz et al. 2016)".*

- L75: What is "rover-based"?

*It is the use of a vehicle as a sensor platform for mobile measurements as for example presented in the cited work of Jakobi et al. (2020). As such wording is rather common in literature, we believe that no addition is needed to the text at this point.*

- L77: I don't see how Dong et al. 2014 and Jakobi et al. 2020 discuss drought and flood events. Consider rephrasing.

*Dong et al. (2014) and Jakobi et al. (2020) do not discuss the topic of the last noun (mentioned in Bogena et al. (2022)) but the topics mentioned in the entire sentence. We proposed the three citations together at the end of the sentence to make the reading easier. We thus believe that no additional changes are needed at this point.*

- L82: "areas with different SM content can be overlooked by a single CRNS" - Please rephrase. Almost all nearby areas do influence the detector signal integratively, so "overlooked" is probably wrong wording.

*We will substitute the previous text with "areas with different SM content can be underrepresented by a single CRNS".*

- L87: "detection of irrigation-related SM variations might not be possible" - Are Li et al. showing that it is not "possible", or are they merely showing that the irrigation signal is within the noise level of typical CRNS detectors? If the latter, there might be chance to detect even small sm changes with better detectors? E.g., higher count rate or improved shielding? I think there is a good chance here to use Li et al. 2019 for motivating your study! Consider rephrasing.

*Following this and the next comments of the reviewer, **this paragraph will be reorganized to present the study of Li et al. (2019) in a single occurrence and better describe the general context. The new text will read:***

*"**Sub-footprint heterogeneity can be reconstructed using multiple instruments, but this comes with increased costs and necessitates further assumptions regarding spatial continuity (Heistermann et al., 2021). As a result, it can be difficult to distinguish local SM variations (Francke et al., 2022), such as the difference between the SM in a small irrigated field and in its surroundings.*

*Despite such limitations, Ragab et al. (2017) reported that CRNS measurements were useful for monitoring soil moisture deficit in the root-zone and Finkenbiner et al. (2019) found that information obtained from combined CRNS measurements and electrical conductivity surveys could improve water use efficiency in a field irrigated with a centre pivot system in Nebraska (USA). Also, Baroni et al. (2018) reported a clear response of CRNS to irrigation, although quantification of single irrigation events was not possible due to effects of precipitation and irrigation of nearby fields.*

*In the case of drip irrigation, where the irrigated area is only a small portion of the volume sensed by the CRNS, the detection of irrigation-related SM variations can be more challenging. For example, in Li et al. (2019), it was not possible to accurately monitor drip irrigation with a standard CRNS in a citrus orchard in Spain. This was a consequence of the relatively small area wetted by drip irrigation, which resulted in a small mean SM change in the instrument footprint. However, better results could be achieved in irrigated fields with a larger wetted area, in drier regions, and for longer and more intense irrigation periods as well as by using instruments with higher count rates".*

- L93: Again Li et al.? Consider merging the two occurances.

*Please refer to the answer provided in the previous comment.*

- L103-106: Three occurances of Köhli et al. 2015 within four lines with mainly identical contexts. Please consider rephrasing.

*This paragraph will be reorganized: "Within this context, the aim of this study is to analyse the feasibility of CRNS-based SM monitoring in irrigated environments. To achieve this, neutron transport and detection in irrigated environments was investigated with physics-based Monte*

*Carlo simulations. These are widely used in CRNS studies (Andreasen et al., 2016) that are focused on, for example, the description of the footprint characteristics (Zreda et al., 2008) and the local site arrangement and instrument calibration strategies (Desilets and Zreda, 2013; Schrön et al., 2017). In this study, the Ultra Rapid Adaptable Neutron-Only Simulation (URANOS) model developed by Köhli et al. (2015) was used".*

- L109: "energy dependent response function" - What is this? Please elaborate in the paragraphs above.

*Here, we refer to the energy sensitivity of the detector, the so-called response function. We will specify this in the new version of the manuscript and add a reference to Köhli et al. (2021).*

- L121: "measure neutrons in the thermal to fast energy regimes" - Can you be more specific or provide a reference?

*We will provide reference to the work of Weimar et al. (2020) and Köhli et al. (2021).*

- L124: Why is Bogena et al. 2022 a good reference for the influence of additional hydrogen pools on CRNS? Consider adding Iwema et al. 2021, doi:10.1002/hyp.14419 and Baroni et al. 2018, doi:10.1016/j.jhydrol.2018.07.053

*We will substitute the original reference with those proposed by the reviewer.*

- L126: "a CRNS" - Here you use CRNS as singular, but the grammar in most of your sentences suggests that CRNS is plural. Please clarify.

*Due to the nature of using only the first letters to form an acronym and to the fact that CRNS is used in literature as an acronym of "cosmic-ray neutron sensor -s" and "cosmic-ray neutron sensing", often in the same manuscript, we decided to not indicate the grammatical number as this is commonly done in literature.*

- L126-132: Unnecessarily detailed on how the detector, the electronics, the meteo station and the antenna work. Consider removing this part from the manuscript (which is about a theoretical, simulated detector).

*We critically considered the comment of the reviewer here. One observation that we would like to stress at this point is that, although purely theoretical in nature, this is one of the first, if not the first manuscript that brings forward the topic of the feasibility of irrigation monitoring with CRNS. Such tool is relatively new in general, and examples of irrigation applications are scarce at best. As such, we expect that many readers will be not familiar with CRNS. Thus, we agree that this section can be shortened but we will not remove it completely as we believe that a short instrument description as well as the references that are provided can represent an added value for readers that are new to the CRNS topic.*

- L138: "The footprint is assumed to be circular" - Schattan et al. 2019 as well as Schrön et al. 2022 showed that it can be asymmetric depending on the site heterogeneity. This

could be also relevant for your study on irrigation, especially if only parts of the outer fields are irrigated. So, is this radial symmetric assumption necessary for your study?

*The reviewer raises a fair point and those recent studies (among others) found that the footprint can be asymmetric. In our study as well, the rectangular shape of the field inevitably leads to minor deviations from a circular shape. This might be kept in mind, yet, as the overall study design focuses on symmetrically aligned topographical elements, such considerations are only of theoretical interest. Only in cases where there are entities of large soil moisture differences close to the sensor and heterogeneously positioned around the sensor, footprint deformations can be of interest. Also, as we stated later in the manuscript, the angular distribution is not subject of this study. We thus believe that the assumption of a circular footprint fits the scope and topic of this study and is the best choice.*

*Nonetheless, to provide the reader with additional information, we decided to extend this topic and include the references proposed by the reviewer. The new text will read: "Although some studies suggested an asymmetric or "amoeba-like" shaped footprint (Schattan et al. 2019; Schrön et al. 2022), most studies assume a circular footprint that depends on the Euclidean distance between the points where neutrons had first contact with the ground and the point of detection."*

- Figure 1: Please indicate the detector position (e.g. with a point) and add scales (in meters) to the inner area box (hectares alone are not very easy to grasp)

*As mentioned in previous comments, we will included the suggested changes in Fig.1.*

- L194-197: These statements sound like results, rather than methods. Or are they already established knowledge? Then please provide references.

*We will include in the text some references to previous works of Weimar et al. (2020) and Köhli et al (2018).*

- L199: Consider adding Rasche et al. 2021, doi:10.5194/hess-25-6547-2021, as a reference for the discussion on thermal neutrons and their behaviour in heterogenous terrain.

*We will include such reference.*

- L201: Why "either"? Can't you use the same neutron for both, count rate calculation and footprint calcuation?

*The previous manuscript version was not fully clear here. The new version will read: "Then, these weights were a) summed to obtain the number of detected neutrons and b) used in a weighted calculation of the R86".*

- L214: As you have mentioned also on other places, consider adding Schrön et al. 2022, as they seem to have demonstrated exactly that.

*We will include such reference.*

- L215: "Here, it is assumed that having a relatively small R86 is beneficial when monitoring irrigation in small fields" - Please rephrase, what does this assumption imply?

*The previous manuscript version was not fully clear here. The new version will read: "Here, the initial hypothesis is that a relatively small R86 is beneficial when monitoring irrigation in small fields as a lower contribution from the surrounding areas could be expected".*

- Eq 1: Please elaborate more on where this equation comes from. It looks like you are propagating the error of N1-N2 (which is the change of neutrons upon detection), where the error of Ni is 1/sqrt(Ni)?

*The reviewer correctly interprets Eq. 1. This is the result of Gaussian error propagation on ND = N1-N2, which means taking the square root of the sum of the squared derivates of ND times its error which is, due to counting statistics, the square root of the resulting number of counts in N1 or N2, respectively. We did not intend to add a long description about basic statistics and thus provided an easier reading. We believe that the original text is sufficient in this case.*

- L232: The cited preprint is not a good reference here. Can you refer to a study which presents typical irrigation intervals?

*We understand the point of the reviewer here as the original text was misleading. We could not find relevant information on detection uncertainty for generic CRNS in evenly irrigated small fields. Thus, we included a generic detection uncertainty for soil moisture monitoring. The new version of the manuscript will clarify this aspect and read: "In addition to σ, a value of α=1% was included in each threshold to represent a generic detection uncertainty limit for a detector that can achieve ~1000 counts per hour and aggregation times of < 12 hours that are relevant in SM monitoring (Schrön et al., 2022)".*

- L248: "inimum and maximum percentages of detected neutrons" - Is this relative to all albedo neutrons, or to all detected neutrons (including non-albedo neutrons)? Please clarify.

*We carefully considered the comment of the reviewer here. We will provide a clear description, and the new text will read: "For each dimension of the simulated inner area and for each moderator type, Table 2 shows the minimum and maximum percentages of detected neutrons that originate in the inner area depending on SM conditions and relative to all detected albedo neutrons".*

- Figure 5: Since panel f) shows a difference rather than the actual value of R86, the colorscale should be completely different. However, it still has blue and yellow colors just like the colormap from the other panels. Please make f) more distinguishable, by choosing a more distinguishable colormap (e.g., red-white-black). Moreover, using terrain colormaps for R86 is not a good choice. Blue stands for water, but low radii have nothing to do with water. Try to use a more linear colormap (e.g., greyscales), or none.

*We carefully considered the comment of the reviewer and critically examined Fig.5 of the original manuscript version. We agree that a simpler and clearer version could be produced. As low R86 can be somehow associated with wet soil conditions and large R86 with dry soil conditions, we decided to use a simpler blue-red scale for the panels a-e. Thus, green (intensity) was a good contrasting choice for panel f. We tested the use of greyscales, but it made the black contours less readable. Also, the use of no colormap resulted in a figure that was like the original version of Fig.3, which we now understand had poor readability. Thus, we believe that this new* Figure 7 *is a good and readable compromise, and we will used it in the new version of the manuscript.*

[Figure]

*Figure 7: alternative figure that substitutes Fig.5 in the new version of the manuscript.*

- Figure 6: Can you add conventional N(sm) functions for the purely homogenous case for comparison?

*Following the reviewer suggestion, we will include relative change in detected neutrons for homogeneous SM variations (i.e., homogeneous area or irrigated field of infinite dimension). The result is shown in the following* Figure 8. *As this new Fig.6 is more informative than the previous version, it will be added to the new version of the manuscript. The caption will be modified accordingly.*

[Figure]

*Figure 8: alternative figure that substitutes Fig.6 in the new version of the manuscript.*

- Figure 8: Why are some bars larger for higher sm than others at lower sm? Is this an effect of the simulation uncertainty? Can you provide an errorbar for the bars?

*The results of this study are subject to statistical uncertainties. When looking at small differences, results are often influenced by fluctuations, which we for example see in Fig.8. In order to quantitatively understand and classify our findings, we provide a measure of certainty by comparing them to the standard deviation of the respective data set. In our opinion, the dashed lines show the error bars of the bars in a clearer and simpler manner. We thank the reviewer here as we also noticed that the red area below σ+α mentioned in the caption of Fig.8 was not visible. We also now use multiple simulation results for the initial homogeneous soil moisture conditions. We will thus provide a new version (Figure 9) of the figure where such area is clearly marked, which will also help the reader in this context. We also will modify part of the caption to have a clearer description: "The bars show the relative change in detected neutrons induced by the irrigation event while the dashed lines show the prescribed detection certainty thresholds.". Please note that further modifications to Fig.8 were made according to the comments of reviewer n.2.*

[Figure]

*Figure 9: alternative figure that substitutes Fig.8 in the new version of the manuscript.*

- Figure B1: % relative to what? Can you add the signal of a completely bare sensor (0 mm HDPE)?

*As mentioned in the caption of the figure, it is "Boxplot of the percentage of detected thermal neutrons over total detected neutrons with different moderator types for the 1 ha scenario" and thus relative to the total number of neutrons detected. We decided to improve the caption which will reads: "Boxplot of the percentage of detected thermal neutrons over the total number of neutrons that are detected with different moderator types for the 1 ha scenario". As the figure shows the fraction of thermal neutrons that are detected and not the count rate, the inclusion of a bare counter to the plot by means of additional investigations would not, in our opinion, provide meaningful additions to the manuscript. According to the general understanding, a thermal counter counts 100 % thermal neutrons. That is, however, not exactly true and the neutron*

*absorption probability in a neutron converter has a 1/sqrt(E) dependence. Given the threshold of thermal energies, the thermal counter would rather lie around 90 % of thermal neutrons. Starting a discussion about why that is the case is not the focus of this publication. Also, we believe that at this point of the manuscript and of the study, the inclusion of a 0 mm HDPE moderator would not provide meaningful additions.*

- The reference list is sorted by first author name, but for the same author it is not sorted by publication year. This makes searching for references very cumbersome, especially for extensivley used author names (such as Bogena et al.). It should be fixed during typesetting.

*This will be improved in the new version of the manuscript.*

- With regard to the previous comment, please double-check whether such a high number of rather general references on soil moisture are necessary for this very specific manuscript about neutron detector simulations and irrigation.

*We will remove the references of those parts of the introduction that will be shortened according to previous comments.*

---

## Author Comment (AC2)

**Response to reviewers**

**Anonymous Reviewer n.2**

This paper investigates the feasibility of CRNS-based SM monitoring in irrigated environments. The paper is informative with lots of simulations in different scenarios. Simulation of neutron count in different scenario was performed with Monte Carlo simulations and Ultra Rapid Adaptable Neutron-Only Simulation. However, obtained results were not validated using real case scenario of irrigated areas.

The author chose a square, not a circle, irrigated area. While the CRNS is tube-shaped and the footprint of the CRNS is circular, is there a reason for that specific shape? Also, do we expect an improved detection rate or minimal if he changed it to circular?

*As Reviewer n.1 also points at the shape of the irrigated field, we provide here the same set of answers. We believe that a circular pivot irrigation field would have not been the best choice for this study. This because circular centre-pivot irrigated fields are very large and typically 400 m in radius (although 500 m are also common and larger exist). Small centre-pivot irrigated fields are not common. The large dimension of a centre-pivot irrigated field means that most of the neutrons detected by a CRNS placed in the middle of such large field (if not all detected neutrons) originate within the irrigated field. Thus, the irrigation that the CRNS would sense is comparable to a rain event, and we think this would not be as interesting as a smaller field of a few ha where the outside area plays an actual role.*

*Relatively small fields as those investigated in this study are most often rectangular and not circular. It is true that the rectangular shape can vary greatly and elongated rectangular shapes are common, but the inclusion of elongated shapes in this study would lead to a much more complex manuscript and would go beyond the scope of the manuscript. Since the manuscript is already long and complex in the current form, we believe that the reader would have strong difficulties in navigating through additional shapes of the irrigated area. Nonetheless, it could be expected that such elongated shapes would be more challenging for the CRNS compared to a squared shape. We agree with the reviewer that the selection of the shape is an important topic, and we will include additional considerations in section 4 "Limitations and outlook". We also think that future research, especially in real-case scenarios, should investigate these aspects and we will mention this in the revised manuscript.*

*Furthermore, we tested the simulation of a circular and rectangular (142x70 m) irrigated fields of 1 ha area and compare the results with those of a squared 1 ha irrigated field. The results are shown in* Figure 1 *in a similar way as they are shown in Fig.8 of the manuscript. In general, differences between the squared, circular, and rectangular shaper are rather small, at least in this simulation setup. Compared to the squared shape, there might be sometimes a small tendency towards higher relative changes in detected neutrons for a circular shape and a tendency towards lower relative changes when a rectangular shape is used. However, results are too similar to draw meaningful conclusions. A clearer picture could possibly be obtained if these two additional scenarios are simulated for the entire soil moisture range of 0.05 to 0.50 cm$^3$ cm$^{-3}$ and for the five investigated areas of the irrigated field. However, this would increase the number of simulations*

*by +200%, which is not feasible due to time and computational constraints and goes beyond the scope of the manuscript. Thus, we decided not to add simulations of different field shapes to the manuscript.*

[Figure]

*Figure 1: CRNS chance of detecting irrigation events of 0.05 and 0.10 cm3 cm-3 (blue and green bars respectively) in a) squared, b) circular, and c) rectangular irrigated field of 1 ha. The bars show the relative change in detected neutrons induced by the irrigation event while the dashed lines show the prescribed detection thresholds. The red area below the σ+α threshold indicates uncertain detection.*

Belo few questions that could make the paper clearer for the reader:

*We have carefully examined the comments provided by the reviewer and we offer here a point-by-point answer.*

Line 14: the unit needs typo correction

*The new version of the manuscript will offer a corrected version for this typo.*

Introduction: Recent work on soil moisture mapping from SAR images is worth reporting in the introduction, primarily those that provide operational soil moisture mapping through the synergistic use of Sentinel-1 and Sentinel-2.

*We agree with the reviewer that such studies on radar-derived SM products are of general interest and could fit the first paragraphs of the introduction. However, reviewer n.1 pointed at the length of the introduction, which will be shortened in the new version of the manuscript, and at the large number of general references on soil moisture. We would thus prefer not to add these further references to the manuscript at this point.*

In line 143-144, you mentioned that variations in humidity, vegetation, and other environmental variables can affect the footprint but with less degree than the SM effect. However, in the simulation, these factors were fixed (line 167-170). Can you explain what the impact would be on the simulation results if you include the diurnal weather conditions?

*We can expect that the diurnal changes in humidity and other environmental variables will affect the CRNS footprint and count rate as mentioned in the manuscript. Regarding the count rate, correction procedures exist for most of such variables and in real-world applications. For example, although atmospheric humidity could vary in an irrigated field, this is typically measured, and the count rate is corrected accordingly.*

*The effect on the footprint, on the other hand, cannot currently be corrected but only explored using neutron transport simulations. Based on this, we can expect a variation in the footprint due to atmospheric humidity changes as shown by Köhli et al. 2015. However, the investigation of a second humidity value would double the quantity of simulations and results. As multiple air humidity values would need to be simulated to obtain meaningful results, we believe that this would result in a too complex picture for the reader and, overall, in a confusing and unfocused manuscript. The same applies to other variables such as vegetation as this would go beyond the scope of the manuscript as the focus is on soil moisture. Nonetheless, we agree with the reviewer that such effects should be mentioned and possibly explored in future research, and we will add considerations on this matter to section "4 Limitations and outlook".*

Line 187: why did you choose 9m radius specifically for all the simulations? Is there a method for choosing the right radius size for the tube?

*The dimension of the virtual detector was set to 9 m as this is commonly done in such simulations with the URANOS model. In selecting the dimension, two aspects should be considered:*
  *a) The smaller the virtual detector, the lower the chance of detecting a simulated neutron and thus the lower the statistical significance of the simulation. This can be counterbalanced by a higher number of simulated neutrons, which however can considerably extend the simulation time and computational needs.*
  *b) A larger virtual detector has higher chance of detection and thus higher statistical significance of the simulations. But the more the dimension of the virtual detector is*

*stretched beyond that of the actual detector, the more secondary effects can influence the*
*simulation results.*
*In the end, a 9 m radius (generally below 5% of the footprint) is a good geometrical limit (and a good compromise between the above-mentioned aspects). This is valid for a typical analysis where there are no variations of the environmental topology in the immediate vicinities of the virtual detector. As these considerations are discussed in some of the literature that the manuscript refers to, we believe that there is no need at this point for additional information on such detail of the methods.*

Looking at Table2 + Figure 3 and then Figure 6, Figure 7 and Fig 9. They are very well connected, I wonder why they did not come directly after each other so that it is easier for the reader to stay on track!

*We agree with the reviewer that such alternative order of the results can offer a nice storyline and a clearer reding. In the new version of the manuscript, we will reorganize the results sections. We will first discuss the footprint dimensions (current Fig.4 and Fig.5) and then the other aspects (Tab. 2, Fig.3, Fig.6, Fig.7, Fig.8, and Fig.9). We will then adapt the consistency of other sections such as abstract, introduction, and conclusions. This will not alter the overall results or the message of the manuscript but will offer a better reading experience.*

Lines 374 – 376: "As shown in Figure 8, an irrigation event that leads to a 0.05 cm3cm-3 increase in SM can be detected with CRNS (relative change in detected neutrons higher than $3\sigma + \alpha$) when the initial SM of the simulation domain is 0.05 cm3cm-3." It looks like there is an overlap in this figure: the green bars represents the 0.05 cm3cm-3 initial SM and the blue bar for the 0.10 cm3 cm-3 . Indeed, the relative change is lower when the increase of soil moisture is higher. Is that correct?

*We thank the reviewer for the comment as we now understand that the readability of Fig.8 should be improved. In Fig.8, the blue bars on the left plots refer to a 0.05 $cm^3$ $cm^{-3}$ irrigation event and the green bars at a 0.10 $cm^3$ $cm^{-3}$ event. The initial soil moisture is indicated on the X-axis at the bottom of the figure. Thus, the relative change is higher when the increase in soil moisture due to irrigation is 0.10 $cm^3$ $cm^{-3}$ (green bars, left side). The relative change is generally lower when the initial soil moisture is higher.*

*To improve the readability of the figure, we will include titles with the irrigation amount on top of the X-axis and we will improve the legend. The new figure will clearly refer to irrigation-related soil moisture changes (see following* Figure 2*). We believe that this new version will be a meaningful improvement, and we thank the reviewer for his comment. Please note that further modifications to Fig.8 were made according to the comments of Reviewer n.1 and due to the current use of multiple simulation results for the homogeneous initial soil moisture conditions.*

[Figure]

Figure 2: alternative figure that substitutes Fig.8 in the new version of the manuscript.

---

## Author Response (AR1)

**Response to reviewers**

*Please find here our response to the two anonymous reviewers. In our opinion, all comments and concerns raised by the reviewers were addressed and this has strengthened the manuscript in a variety of points. We hope that this revised version of the manuscript can be accepted for publication in Geoscientific Instrumentation, Methods and Data Systems.*

**Response to Anonymous Reviewer n.1**

This study investigates the feasibility of soil moisture (sm) detection with cosmic-ray neutron sensors (CRNS) in the center of a squared irrigated field. The authors investigate the sm-neutron relationship and the footprint radius for various scenarios of detector shielding material, field sizes, and combinations of soil moisture conditions in the inner and the outer areas. They conclude that different thickness of HDPE material influences the footprint radius and the detector sensitivity, with HDPE 25mm + Gadolinium leading to the highest sensitivity to soil moisture changes and to the largest footprint radius. Moreover, they found that soil moisture from the outer fields substantially influences the CRNS sensitivity to the irrigated field, especially for small fields.

The study is very well written and highly relevant, being the first of its kind to study various detector thicknesses and gadolinium shields with regards to the neutron-soil moisture relationship. This will be a very important information for CRNS users and suppliers in order to significantly improve the sensor performance. Furthermore, the investigation of irrigation support is also very relevant for current and future CRNS applications. However, this study is only theoretical and limited to a very specific geometry (CRNS within a squared irrigated field). Nevertheless, it is a good step forward.

Major concerns with this study should be clarified before publication of this manuscript, particularly regarding the model assumptions, the consistency of the argumentation, and the integrity and applicability of the conclusions. Please address also the minor comments and the specific comments below.

*We sincerely thank the reviewer for the appreciation of our work and for the time spent to produce a thorough and useful review. We have carefully considered all comments and observations. We provide here replies to each point with indications on the changes that were included in the revised version of the manuscript or motivations for not applying further edits when this was not deemed necessary.*

**Major concerns:**
1.  The authors seem to change the water content of the irrigated soil in the whole soil column (up to 160 cm depth). However, it is highly unrealistic to assume vertically homogenous wetting of the soil due to surface irrigation! Hence, I wonder how useful the results of this study are for practical applications. In the real world, surface irrigation adds moisture only to the first few centimeters of the soil, or decimeters, depending on typical depth of the roots and the hoses. This would leave a large part of the vertical footprint dry, such that the

results obtained in this study might significanlty overestimate the actual sensitivity of CRNS to irrigated soil.

*We thank the reviewer for this observation since, although known to the authors, this limitation was only briefly mentioned in Chapter "4 Limitations and outlook". We agree that the wetting due to precipitation or irrigation is not homogeneous with depth. The wetting caused by irrigation would most probably start at the soil surface and reach the deeper soils within a certain amount of time depending on several variables (see below). Typically, we can expect deeper soils to have a higher SM compared to the soil surface when irrigation starts. Then, SM would increase in the soil column and peak at the top of the profile. This is followed by a redistribution of SM that can take up to several days/weeks.*

*We assumed a homogeneous SM distribution for our study because we considered that SM dynamics in soils are a rather complicated subject. The SM distribution in the soil column before, during, and after an irrigation event can depend on (among other factors):*
  *- Soil texture, porosity, bulk density, and presence of biopores.*
  *- Previous SM conditions within the soil profile and depth of the water table.*
  *- Root depth and distribution, vegetation type, and root water uptake.*
  *- Agricultural management (ploughing depth and irrigation type).*
*Thus, to define how the SM varies in depth in an agricultural area, it would be necessary to consider at least soil depth and texture, water table depth, ploughing depth, rooting depth, and root distribution.*

*Another aspect to consider is the variation of the depth of investigation of a CRNS with SM variations as well as with increasing distance from the instrument (larger below the instrument and relatively shallower at distance). Thus, the overestimation of the contribution of irrigation to the signal would be present mostly in the vicinity of the instrument. Also, during an irrigation event, the wetting of the upper soil would reduce the depth of investigation of the CRNS and thus diminish the relative contribution of deeper soils to the signal. We believe that these two factors could reduce, although not completely remove, the overestimation mentioned by the reviewer.*

*Overall, an investigation of the influence of vertical heterogeneity would add a further dimension to the problem that is investigated in this study and would significantly increase its complexity. As it is not feasible to include all possible scenarios into the simulation of a CRNS framework, this theoretical study presents one particular solution to one subset of problems, which we believe is already of great interest. Further research is necessary (and currently carried out by some research groups) to integrate vertical heterogeneity based, for example, on hydrological models. But this again is, in our opinion, outside the scope of this study.*

*This said, the observations of the reviewer are valid, and we agree that we did not discuss this specific limitation in a sufficient way in the previous version of the manuscript. In the revised version of the manuscript, we point at this limitation in Chapter "4 Limitations and outlook" where the revised text reads "Moreover, the possible influence of additional environmental conditions such as vertical SM distribution, atmospheric variables (e.g., humidity), and vegetation was not considered here. For example, a heterogeneous vertical SM distribution may result in a reduction*

*in performance of the CRNS if only the topsoil is wetted by an irrigation event". Also, future research that could investigate such topic is mentioned in the Conclusions.*

2. The authors conclude that 25 mm HDPE+Gd is the best shielding variant for CRNS detectors (L359) without studying other HDPE thicknesses together with Gd. It is indeed interesting to see gadolinium used for the first time in a sophisticated analysis of detector sensitivity to footprint and soil moisture changes. However, the authors used Gd only in the 25 mm HDPE setup. In order to distinguish and better understand the effects of HDPE and Gd on the variables of interest, and to find the best performing detector design, it would be necessary to simulate Gd shielding also for different HDPE thickness.

*We agree with the reviewer that investigating the effects of a Gadolinium based shielding on different moderator thicknesses would be interesting, at least in general terms. However, to our knowledge, such shielding is currently commercialized only with 25 mm HDPE moderators. Additionally, as shown in the following* Figure R 1*, a Gadolinium based shielding prevents the detection of approximately 90% of thermal neutrons. This effect differs from a HDPE moderator since the sensitivity is not shifted towards higher energies but rather cut in the thermal range. Thus, it can be expected that a 25 mm HDPE moderator with Gadolinium shielding would generally detect less neutrons than a 25 mm HDPE moderator without Gadolinium shielding.*

[Figure]

*Figure R 1: Response functions with the thermal neutron energy range marked with a red area.*

*We could then expect that, by adding Gadolinium shielding to a thin moderator that has high sensitivity to thermal neutrons and low sensitivity to epithermal and fast neutrons (e.g., 5 mm HDPE), the number of detected neutrons would decrease considerably (partial removal of the red shaded area in* Figure R 1*). In thicker moderators, the work of Weimar et al. (2020) shows that for moderator thicknesses of 20, 22.5, 25, and 27.5 mm HDPE there are only small differences when the Gadolinium shielding is added. The authors concluded that a 22.5 m HDPE moderator with a Gadolinium thermal shielding provided the best overall performance but with only a marginal difference compared to a 25 mm HDPE moderator (as proposed by Desilets et al. (2010)) with Gadolinium shielding. It is true that thinner moderators yield slightly better signal dynamics at the cost of a small reduction of the count rate. However, such differences are marginal and, in our opinion, not significant when put in the framework of a study such as the one proposed here.*

*In the end, to our knowledge, a 25 mm HDPE moderator with Gadolinium shielding is the common choice because of a) recommendations from, for example, Desilets et al. (2010) confirmed by Weimar et al. (2020) and b) production costs. Previous research has shown the influence of different moderator thicknesses, at least in a range of moderators that is relevant. We thus believe that further investigations on this topic would go out of the scope of the study and not add novel information for the reader.*

*We now elaborate on the choice of a 25 mm HDPE moderator with Gadolinium shielding in Chapter 2.5 where the revised text reads: "Furthermore, an additional gadolinium oxide (Gd2O3) shield was investigated for a moderator composed of 25 mm HDPE. For the gadolinium variant, a 25 mm HDPE moderator thickness was selected as it was proposed by Desilets et al. (2010). In addition, Weimar et al. (2020) found only small differences in response for moderator thicknesses of 20 to 27.5 mm HDPE with gadolinium shielding. Thus, the combination of gadolinium shielding with other moderator thicknesses was not considered."*

3. The authors conclude that outer sm is an important factor for irrigation monitoring with CRNS by considering only one shielding option that provides the largest footprint. In section 3.4 and 3.5 the impact for inner sm changes on the neutron detector sensitivity is investigated, but only for 25 mm HDPE-Gd (mentioned in 3.4, not mentioned in 3.5). In the same study, the authors find that different shielding options have different footprint radii. So, I would expect that shielding options with lower footprint radii would lead to less impact of the outer sm. Please elaborate on this and whether different shielding options could be recommended for smaller fields than for larger fields to reduce the influence. This might change your conclusion. Also, use this opportunity to join the two otherwise separate parts of "shielding/footprint analysis" and "sm/area analysis", in order to demonstrate that a combination of those two studies in a single paper could actually lead to a greater benefit.

*In the revised version of the manuscript, it is clearly stated in Chapter 3.5 that the analysis is performed for a 25 mm HDPE moderator with Gadolinium Shielding. This was not the case in the previous version of the manuscript, and we thank the reviewer for highlighting it.*

*It is true that the Gadolinium shielding provided a footprint that is larger than that of the other moderators. However, it should be noted that the R86 of the 25 mm HDPE moderator with Gadolinium shielding was approximately 5 to 15 m larger than that of the other moderators (with R86 values between ~120 and ~270 m, see Figure 3 of the revised manuscript). Here, we noted that we did not clearly state in the previous version of the manuscript that such differences in R86 are rather small and this is now mentioned in the revised version of the manuscript.*

*Additionally, in Figure 6 of the manuscript, we show how the 25 mm HDPE with Gadolinium shielding offers a higher sensitivity compared to the other moderators. Thus, although a different moderator with a smaller R86 (e.g., 5 mm HDPE) could possibly lead to a reduced influence of the SM outside the investigated field, the moderator with the highest sensitivity (25 mm HDPE with Gadolinium shielding) would still be preferable. Therefore, we decided to focus the second part of the results on such moderator only. In Chapters 3.4 and 3.5, we could have included results from all eight moderators, and we initially tried to do this. However, we soon realized that this*

*resulted in an overwhelming and confusing amount of information with no real benefit for the overall message of the study.*

*We nonetheless think that the reviewer is raising a fair point and decided to provide additional figures in our reply to the comments. The following* Figure R 2 *was produced starting from Figure 9 of the original manuscript (Part-i). To this, we added similar figures obtained with data from a 25 mm HDPE moderator (Part-ii) and 5 mm HDPE moderator (Part-iii).*

[Figure]

*Figure R 2: Part-i is a copy of Figure 9 of the manuscript (referring to a 25 mm HDPE moderator with Gadolinium shielding). Part-ii and Part-iii are figures similar to Figure 9 of the manuscript built with data from a 25 mm HDPE and 5 mm HDPE moderators respectively.*

*As visible in* Figure R 2, *the use of a 25 mm HDPE moderator without Gadolinium shielding, which has an R86 that is 5-10 m smaller than the version with Gadolinium, does not show a clear reduction of the influence of SM outside the irrigated field. However, there is a general reduction in sensitivity, as can also be seen in Figure 6 of the manuscript. Furthermore, a 5 mm HDPE moderator, although having a footprint that is up to 20 m smaller than that of the 25 mm HDPE with Gadolinium variant, also does not show a noticeable reduction of the effect of the outside SM. In fact, the effect of the outside SM appears higher than in the Gadolinium case. However, the most important difference is the generally lower sensitivity of the moderator without gadolinium.*

*We believe that this could be connected to the fact that the detected neutron intensity of a CRNS peaks nearby the instrument and decreases with distance. Thus, a small variation in the R86 alone might not lead to meaningful differences between moderators and the instrument with higher sensitivity to SM changes should be preferred. We now realize that this concept was not fully described throughout the original manuscript, and we point at this aspect in Chapter 3 "Results and Discussion" as well as in the Conclusions. For example, in the latter, the revised text reads: "Generally, detectors with thinner HDPE moderators result in smaller footprints although differences between the investigated moderators are relatively small" and "Despite a relatively larger R86 combined with lower contributions from the irrigated field, thicker HDPE moderators and the addition of a thermal shielding result in higher relative changes in detected neutrons with respect to SM variations. Thus, such moderator types are expected to bring improvements in CRNS-based irrigation monitoring". Also, the results of* Figure R 2 *are summarized and included as an appendix (Appendix C) in the revised manuscript.*

4. The authors suggest that sm measurements outside the irrigated area are needed to properly estimate sm inside with CRNS. But in this case, what is the added value of CRNS compared to using the suggested number of additional sensors and putting them inside from the start?

*We carefully considered the comment of the reviewer and we realized that we did not offer enough information in the previous version of the manuscript. What we suggest is the installation of a SM sensor outside the irrigated field or, alternatively, the use of a portable device. This scenario of CRNS plus a single device is different compared to the use of a sensor network within the irrigated field as a) only the CRNS would be installed in the field, b) portable or point-scale SM measurement devices are relatively inexpensive, and c) measurements of the SM outside the irrigated field could provide correction for multiple CRNS installed in fields that are located within a large area. The revised version of the manuscript now briefly mentions these considerations in Chapter 4 by including this statement: "For this, a CRNS in an irrigated field could be supported by a single and inexpensive point-scale SM monitoring instrument installed outside the target field. Such addition would not substantially increase the installation and maintenance costs and would not interfere with agricultural management. Moreover, a single point-scale device could possibly support multiple CRNS in an agricultural area if irrigated fields are sufficiently distanced and if the SM in the unmanaged area is relatively homogeneous in space".*

5. The study is limited to one specific case of field geometry: a CRNS detector in the center of a square-shaped irrigated field surrounded by a homogeneous field on all sides. Although simulations of various shielding types and square sizes are scientifically

interesting, the transferability of the results to practical field geometries might be very limited. I would recommend to better communicate this limitation, at first by improving Figure 1 with a detector symbol in the center and scales in meters. Second, by providing real-world examples where this scenario could be applicable (I would rather think that radial field geometries would have been a better choice, e.g. for pivot irrigation). Third, by discussing potential deviations and uncertainties of the results if the sensor location would not be ideally centric, or if the field shape is a circle instead of a square. This would allow users to get a better idea whether the results would be still applicable to a certain degree, or whether completely new simulations would be necessary for every single deviation from the presented ideal case.

*We included the suggested changes in Figure 1 (see following* Figure R 3*) and, as described in detail below, we added considerations on the selected shape of the irrigated field in Chapter 4 "Limitations and outlook".*

[Figure]

*Figure R 3: alternative figure that substitutes Figure 1 in the revised version of the manuscript.*

*Reviewer n.2 also points at the shape of the irrigated field. We believe that a circular pivot irrigation field would not have been the best choice for this study because circular centre-pivot irrigated fields are very large and typically 400 m in radius (although 500 m systems are also common and larger systems exist). Small centre-pivot irrigated fields are not common. The large dimension of a centre-pivot irrigated field means that most of the neutrons detected by a CRNS placed in the middle of the field originate from within the irrigated field. Thus, the irrigation that*

*the CRNS would sense is comparable to a rain event, and we think this would not be as interesting as a smaller field of a few ha where the outside area needs to be considered.*

*Relatively small fields as those investigated in this study are most often rectangular and not circular. It is true that the rectangular shape can vary greatly and elongated rectangular shapes are common, but the inclusion of elongated shapes in this study would lead to a much more complex manuscript and would go beyond the scope of the manuscript. Since the manuscript is already long and complex in the current form, we believe that the reader would have strong difficulties in navigating through additional shapes of the irrigated area. Nonetheless, it could be expected that such elongated shapes would be more challenging for the CRNS compared to a squared shape. We agree with the reviewer that the selection of the shape is an important topic, and we include additional considerations in Chapter "4 Limitations and outlook". The revised text reads: "Different dimensions of the irrigated field, such as irregular or elongated shapes, might be more challenging for irrigation monitoring with CRNS.". We also think that future research, especially in real-world scenarios, should investigate these aspects and we mention this in Chapter 4 as well.*

*To obtain first insights into the effects of field shape, we performed additional simulations of a circular and rectangular (142x70 m) irrigated field of 1 ha area and compared the results with those of a square 1 ha irrigated field. The results are shown in* Figure R 4 *in a similar way as they are shown in Figure 8 of the manuscript. In general, differences between the squared, circular, and rectangular shaper are rather small, at least in this simulation setup. Compared to the square shape, there might be sometimes a small tendency towards higher relative changes in detected neutrons for a circular shape and a tendency towards lower relative changes when a rectangular shape is used. However, results are too similar to draw meaningful conclusions. A clearer picture could possibly be obtained if these two additional scenarios are simulated for the entire soil moisture range of 0.05 to 0.50 cm3 cm-3 and for the five investigated areas of the irrigated field. However, this would increase the number of simulations by +200%, which is not feasible due to time and computational constraints and goes beyond the scope of the manuscript. Thus, we decided not to add simulations of different field shapes to the manuscript.*

[Figure]

*Figure R 4: CRNS chance of detecting irrigation events of 0.05 and 0.10 cm3 cm-3 (blue and green bars respectively) in a) squared, b) circular, and c) rectangular irrigated field of 1 ha. The bars show the relative change in detected neutrons induced by the irrigation event while the dashed lines show the prescribed detection thresholds. The red area below the σ+α threshold indicates uncertain detection.*

*Regarding the positioning of the CRNS, we think that the assumption that the instrument is positioned in the centre of an irrigated field is a common practice. However, CRNS are sometimes not positioned in the centre of a field, for example, when multiple CRNS are placed in different areas of a single field. But we believe that the discussion and investigation of device placement that are not in the centre of a given field (or its vicinities) as well as the placement of multiple devices goes beyond the scope of this study. Regarding the need for new simulations prior to CRNS installation, we suggest in the original version of the manuscript (Chapter 4 "Limitations and outlook" and conclusions) that neutron transport simulations could be useful to assess costs and benefits before CRNS installation as well as to quantify the partitioning of neutron origins in real case scenarios. We thus do not believe that these simulations are currently exhaustive, and we*

*think that we did not suggest their exhaustion in the manuscript. We nonetheless revised the text in Chapter 4 "Limitations and outlook" to better communicate these topics.*

**Minor comments**

1. The authors mention that sm change outside can be larger than sm change from irrigation inside. However, it is not clear in which scenarios it is actually realistic to assume that sm changes "only" outside. Most of the time, precipitation occurs within CRNS footprints rather homogeneously. So the only cases I could image of sm changing outside (and not inside) is that these fields are also irrigated, and with a completely different schedule. You could communicate more openly that this is a special case, and only in this case the sm-outside issue becomes relevant.

*We agree with the reviewer, and we understand now that we created some degree of confusion in certain instances. The revised version of the manuscript clarifies this aspect at the beginning of Chapter 3.5 where the irrigation of neighbouring fields is now mentioned.*

2. The authors seem to assume that only neutrons that originated from the inner area carry its sm signal (L211). However, Köhli et al. 2015 mentioned a few intermediate interactions of neutrons from longer distances with the soil on their way to the detector. In this context, it seems that neutrons from outer regions could carry signals from the inner regions, too. In this case the authors are encouraged to reconsider their assumption.

*We now understand that we presented this assumption in a rather simplified way. In the revised version of the manuscript, we clarify that the neutrons that originate within the irrigated area carry a large portion of the information of interest. In our opinion, this does not change the general message that a large percentage of detected neutrons that originate outside the irrigated field is a challenge in irrigation monitoring with CRNS.*

*In the revised version of the manuscript, we extended Chapter 2.6 which now reads "Here, it was assumed that the neutrons that originate within the inner irrigated field carry the bulk of the information of interest. Although neutrons that originate outside the irrigated field can have occasional within-field interactions before reaching the CRNS (Köhli et al. 2015), these were considered of secondary importance for the scope of this study". Similarly, the revised Chapter 3.2 reads: "This percentage represents the detected neutrons that originate within an irrigated field and thus carry the bulk of the information of interest in case of irrigation applications".*

3. Minimum soil moisture used in this study is 0.05 m³/m³, while irrigation might be particularly interesting in extremely arid regions, where sm below 0.05 can exist. Given the very steep neutrons-sm function, there is a potentially significant performance increase of CRNS for dry soils. Can you add ~0.03 m³/m³ to your analysis?

*We carefully considered the possibility to add 0.03 m³/m³ to our simulations. However, we finally concluded that this would not add important information or change the message and conclusions of the manuscript. Although it is true that soils can theoretically have a SM content below 0.05 m³/m³ (e.g., residual soil moisture), we believe this value is very unlikely in agricultural fields*

*(especially for irrigated fields). Thus, we believe that an average SM of 0.05% in the whole soil column is already a sufficiently low SM value for the scope of this study.*

*Regarding the results that we could expect by including a SM of 0.03 cm³ cm⁻³, (see, for example, Figure 8 of the revised manuscript) an irrigation event that leads to a final SM of 0.08 cm³ cm⁻³ (plus 0.05 cm³ cm⁻³ as done for the other SM cases) would be easily sensed by the CRNS since the SM variation 0.05 to 0.10 cm³ cm⁻³ is already sensed. The same can be expected for a 0.10 cm³ cm⁻³ SM variation (from 0.03 to 0.13 cm³ cm⁻³). At the same time, we do not think that it is necessary to investigate a SM increase from 0.03 cm³ cm⁻³ to 0.05 cm³ cm⁻³ as this does not seem relevant in irrigation.*

4. Nomograms for the presentation of the results are very hard to read (e.g. Fig 3). I can understand the authors idea to put both, the inner and the outer sm on the two axes, but it took me several minutes starring at the plot to understand what they are showing. And now that I understand it, I still find it hard to read out what fraction of neutrons comes from the inner or the outer part for a given soil moisture condition. Especially since both relative neutrons are not adding up to 100% (due to direct neutrons?). So, I would strongly suggest to reconsider these graphs, focusing on the main message, which probably is: "How do neutrons from inner and outer areas compare?". If two variables need to be compared, try to show them in the same plot. And since they add up to a total neutron count (or to 100% inlcuding direct neutrons), the usage of stacked barplots might be good choice. One advantage of a N-over-SM plot would also be that the curves could be easily compared with the conventional N(SM) functions to show how this function changes for different irrigation pattern. Just like in Figure 9. Consider replacing Fig 3 with Fig 9 or at least refering to it.

*We understand now that the readability of Figure 3 (Figure 5 in the revised manuscript) can be improved. We also agree with the reviewer that clear indication on the presence of non-albedo neutrons in the calculation should have been included in the description of the figure.*

*For the revised version of the manuscript, we first attempted to create a figure based on stacked bar plots as this was one of the suggestions from the reviewer. The result can be seen in the following Figure R 5. Although some details of the figure could be improved, we think that this illustration method does not allow to clearly show many simulation results. Although the difference between the 0.5 and the 8 ha cases is crystal clear (left and right column respectively), the differences in neutron origin between the 100 simulations for a 0.5 ha field is hardly readable. The same applies to the 8 ha simulations.*

[Figure]

*Figure R 5: Alternative to Figure 3 (Figure 5 in the revised manuscript) of the original manuscript using stacked bar plots. Not selected.*

*Nonetheless, we still agree with the reviewer comment on this figure and tried an alternative illustration. We followed the suggestion of using Figure 9 style and the result is the following Figure R 6.*

[Figure]

*Figure R 6: alternative figure that substitutes Figure 3 in the revised version of the manuscript.*

*In this figure, we decided to not show the results of all simulations as the plotted lines would sometimes get too close to each other. Thus, we only show results for three values of SM outside the irrigated field, which we believe is sufficient. We also show only detected albedo neutrons now, and thus the percentages sum is 100%. We think that this new figure (Figure 5 in the revised manuscript) is more easily readable, and it conveys the messages that are described in the text.*

5. Parts of the introduction are unnecessary or not clear. There are long paragraphs about food security and irrigation in general, point-to-large scale sensors, CRNS detectors in general and their multifaceted applications from snow to regional modeling, and so forth. All this sounds like a great literature review, but it is out of scope in large parts. Instead, in the end of the introduction the very important concept of "energy dependent response function" is just mentioned, while it has never been introduced. Please consider shorteing the introduction and provide a more concise structure with a focus on the actual topic: simulation of multiple neutron detector variants in heterogeneous irrigated terrain. Unclear: the argumentation about thermal neutrons, how are they different from epithermal neutrons, and why is it necessary to exclude them? Here would be a good spot to elaborate on energy response functions. See also the 16 specific comments below.

*We shortened and merged the first two paragraphs of the introduction to have a better focus on the topic of the manuscript. Although we strongly considered the comment of the reviewer, one observation that we would like to stress at this point is that, although purely theoretical in nature, this is one of the first, if not the first manuscript that brings forward the topic of the feasibility of*

*irrigation monitoring with CRNS. Such tool is relatively new in general, and examples of irrigation applications are scarce at best. As such, we expect that many readers will not bet familiar with CRNS and their applications. Thus, we agree that the introduction can be shortened but we did not completely remove certain paragraphs as we believe that they can represent an added value for readers that are new to the CRNS topic. Finally, we now better introduce and describe certain topics such as the difference between thermal and epithermal neutron detection. The specific comments provided by the reviewer were also addressed as described in the following.*

**Specific comments:**
- L47: I don't see how Andreasen et al. 2016 demonstrates that CRNS can close gap between point and large scales. Please double-check the reference and think about providing references related to the footprint and in comparison with actual point measurements (for example, Heistermann et al. 2021, doi:10.5194/hess-25-4807-2021)

*The original reference was substituted with the one proposed by the Reviewer.*

- L49: "cosmic-ray radiation" is tautologous.

*In this case, "cosmic-ray radiation" is not tautologous as it refers to radiation induced by cosmic radiation. Cosmic rays are (primarily) high-energy protons. By interaction with the atmosphere, they create showers of secondary particles, which are, in case of neutrons, a different type of radiation. Therefore, the term "cosmic-ray induced neutrons" is often used instead of the shorthand versions "cosmic-ray neutrons" or the mentioned "cosmic-ray radiation". We nonetheless modified the text in "cosmic radiation" to have it consistent with the Materials and Methods Chapter.*

- L50: "The detected neutrons are generally in the thermal ... or epithermal ...". Please rephrase to avoid questions like: What means "generally"? Can a detector directly detect epithermal neutrons? Is a spectrometer involved? Can you provide a reference for the energy sensitivity?

*In the introduction, we provide a simplification of the actual processes that are involved in neutron detection, both technically and physically. In Weimar et al. (2020) the principles of neutron detection for a CRNS instrument are described at length. After considering the comment of the reviewer, we decided to add a reference to Weimar et al. (2020) in this sentence. Specifically, the largest part of detected neutrons lies within the given energy ranges. More importantly, most of the details are included in the simulation and therefore do not alter the results. Other details, which are not represented in the simulations are, for example, the actual geometry of the sensor. Francke et al. (2022) however, show that the differences between a simulated virtual detector and a simulated detector with its actual dimensions and materials are rather small. We believe that the text is sufficiently informative here after the addition of the abovementioned reference.*

- L52: Consider adding a references that is less than 10 years old, e.g., Köhli et al. 2021, doi:10.3389/frwa.2020.544847

*The reference was added to the revised text.*

- L56: "dry soils have a higher neutron density" - Please rephrase. Is it the soil that "has" this neuton density, or the atomic nuclei inside the silicate atoms? Or the air above the soil?

*We modified the text as follows: "For example, the count rate is inversely proportional to SM since dry soils result in higher environmental neutron density that allows more accurate measurements compared to wet soils".*

- L49-57: The whole paragraph: you start with telling that CRNS is both, thermal and epithermal. Are all these statemens in the text (sensitivity to sm, snow, vegetation, ...) refererring to thermal, to epithermal, or both? Do they behave equally? If no, how are thermal neutrons different? If yes, why do you want to exclude them?

*The comment of the reviewer resulted in a rewriting of the paragraph which now reads: "Additionally, neutrons detected by CRNS are generally in the thermal (below 0.5 eV) or epithermal (0.5 eV to 0.5 MeV) energy regime (Weimar et al., 2020), with the former having smaller footprint and penetration depth as well as different sensitivity to SM and biomass (Jakobi et al., 2022; Jakobi et al., 2021). To enhance the detection of epithermal neutrons, the energy sensitivity (i.e., energy dependent response function) of a CRNS (Köhli et al., 2021) can be shifted towards the epithermal energy range by using a high-density polyethylene (HDPE) moderator (Weimar et al., 2020; Desilets et al., 2010). In addition, a gadolinium-based (Ney et al., 2021) or cadmium-based (Andreasen et al., 2016) shielding can be used to prevent the detection of thermal neutrons (Desilets et al., 2010)".*

*We believe that this revised text addresses the comment of the reviewer.*

- L60: I don't see how Schrön et al. 2017 uses multiple counter tubes to achieve higher count rates. Do you mean Schrön et al. 2018, doi:10.5194/gi-7-83-2018?

*We substituted the original reference with the proposed one.*

- L64: "... prevent the detection of thermal neutrons" - Not clear: why should the detection of thermal neutrons be prevented? Please restructure this paragraph to explain why thermal neutrons behave differently and why they are not useful for soil moisture monitoring.

*For the answer to this comment, please refer to the above comment to L47-57.*

- L69-70: You argue that CRNS does not need to be removed during harvest. Is the instrument of infinitesimal size? If not, what is the spatial extent of the apparature including anchoring, and how can a farmer pull around it? Please consider reporting about pros and cons of CRNS more objectively.

*The previous version of the manuscript described this part in a rather simplistic way, which may be interpreted as an understating of the CRNS disadvantages. Indeed, the instrument is not infinitesimal and occupies a space of approximately 35x35 to 80x80 cm, depending on the*

*instrument type, manufacturer, and installation. Our goal was to highlight how a single CRNS differs from a sensor network composed of multiple distributed nodes. In this case, the farmer needs to drive around a single sensor (generally well visible and placed in the middle of the field) instead of driving around multiple sensors that are distributed within the field. In the latter case, the complication of driving around multiple sensors generally results in the need for complete removal and reinstallation. This is not the case with a single CRNS. We thus improved the text in the revised version of the manuscript by including additional wording from Franz et al. (2016): "In the context of agricultural applications, a CRNS can be placed in between or out of the way of routine production practices. It consequently does not present the logistic challenges associated with directly inserted sensors, which need to be removed and reinstalled during harvest, planting, and other management actions (Franz et al. 2016)".*

- L75: What is "rover-based"?

*It is the use of a vehicle as a sensor platform for mobile measurements as for example presented in the cited work of Jakobi et al. (2020). As such wording is rather common in literature, we believe that no addition is needed to the text at this point.*

- L77: I don't see how Dong et al. 2014 and Jakobi et al. 2020 discuss drought and flood events. Consider rephrasing.

*These references were separated to better reflect this sentence.*

- L82: "areas with different SM content can be overlooked by a single CRNS" - Please rephrase. Almost all nearby areas do influence the detector signal integratively, so "overlooked" is probably wrong wording.

*We substituted the previous text with "areas with different SM content can be wrongly represented by a single CRNS".*

- L87: "detection of irrigation-related SM variations might not be possible" - Are Li et al. showing that it is not "possible", or are they merely showing that the irrigation signal is within the noise level of typical CRNS detectors? If the latter, there might be chance to detect even small sm changes with better detectors? E.g., higher count rate or improved shielding? I think there is a good chance here to use Li et al. 2019 for motivating your study! Consider rephrasing.

*Following this and the next comments of the reviewer, this paragraph was reorganized to present the study of Li et al. (2019) in a single occurrence and better describe the general context. The revised text reads: "Sub-footprint heterogeneity can be reconstructed using multiple instruments, but this comes with increased costs and necessitates further assumptions regarding spatial continuity (Heistermann et al., 2021). As a result, it can be difficult to distinguish local SM variations (Francke et al., 2022), such as the difference between the SM in a small, irrigated field and its surroundings. Despite such limitations, Ragab et al. (2017) reported that CRNS measurements were useful for monitoring soil moisture deficit in the root-zone and Finkenbiner et al. (2019) found that information obtained from combined CRNS measurements and electrical*

*conductivity surveys could improve water use efficiency in a field irrigated with a centre pivot system in Nebraska (USA). In addition, Baroni et al. (2018) reported a clear response of CRNS to irrigation, although quantification of single irrigation events was not possible due to effects of precipitation and irrigation of nearby fields. In the case of drip irrigation, where the irrigated area is only a small portion of the volume sensed by the CRNS, the detection of irrigation-related SM variations can be more challenging. In Li et al. (2019), it was not possible to accurately monitor drip irrigation with a standard CRNS in a citrus orchard in Spain. This was a consequence of the relatively small area wetted by drip irrigation, which resulted in a small mean SM change in the instrument footprint. However, better results could be achieved in irrigated fields with a larger wetted area, in drier regions, and for longer and more intense irrigation periods as well as by using instruments with higher count rates".*

- L93: Again Li et al.? Consider merging the two occurances.

*Please refer to the answer provided in the previous comment.*

- L103-106: Three occurances of Köhli et al. 2015 within four lines with mainly identical contexts. Please consider rephrasing.

*This paragraph was reorganized as follows: "Within this context, the aim of this study is to analyse the feasibility of CRNS-based SM monitoring in irrigated environments. To achieve this, neutron transport and detection in irrigated environments was investigated with physics-based Monte Carlo simulations. These are widely used in CRNS studies (Andreasen et al., 2016) that are focused on, for example, the description of the footprint characteristics (Zreda et al., 2008) and the local site arrangement and instrument calibration strategies (Desilets and Zreda, 2013; Schrön et al., 2017). In this study, the Ultra Rapid Adaptable Neutron-Only Simulation (URANOS) model developed by Köhli et al. (2015) was used".*

- L109: "energy dependent response function" - What is this? Please elaborate in the paragraphs above.

*Here, we refer to the energy sensitivity of the detector, the so-called response function. This is now better specified in the revised version of the manuscript, and we also added a reference to Köhli et al. (2021).*

- L121: "measure neutrons in the thermal to fast energy regimes" - Can you be more specific or provide a reference?

*We provided reference to the work of Weimar et al. (2020) and Köhli et al. (2021).*

- L124: Why is Bogena et al. 2022 a good reference for the influence of additional hydrogen pools on CRNS? Consider adding Iwema et al. 2021, doi:10.1002/hyp.14419 and Baroni et al. 2018, doi:10.1016/j.jhydrol.2018.07.053

*We substituted the original reference with those proposed by the reviewer.*

- L126: "a CRNS" - Here you use CRNS as singular, but the grammar in most of your sentences suggests that CRNS is plural. Please clarify.

*Due to the nature of using only the first letters to form an acronym and to the fact that CRNS is used in literature as an acronym of "cosmic-ray neutron sensor -s" and "cosmic-ray neutron sensing", often in the same manuscript, we decided to not indicate the grammatical number as this is commonly done in literature.*

- L126-132: Unnecessarily detailed on how the detector, the electronics, the meteo station and the antenna work. Consider removing this part from the manuscript (which is about a theoretical, simulated detector).

*We critically considered the comment of the reviewer here. This part was substantially shortened although a few key details and the references to descriptions of CRNS detectors were kept. This because, although purely theoretical in nature, this is one of the first, if not the first manuscript that brings forward the topic of the feasibility of irrigation monitoring with CRNS. As such, we expect that many readers will be not familiar with CRNS detectors. Thus, we substantially shortened but not completely removed this part of the text as we believe that it can represent an added value for readers that are new to the CRNS topic. The revised text reads: "Generally, a CRNS is composed of one or more neutron detectors that can be bare (thermal-epithermal neutron detection) or moderated with HDPE (epithermal to fast neutron detection). More detailed information on the main detector components and physics can be found in Zreda et al. (2012), Schrön et al. (2018b), and Weimar et al. (2020)".*

- L138: "The footprint is assumed to be circular" - Schattan et al. 2019 as well as Schrön et al. 2022 showed that it can be asymmetric depending on the site heterogeneity. This could be also relevant for your study on irrigation, especially if only parts of the outer fields are irrigated. So, is this radial symmetric assumption necessary for your study?

*The reviewer raises a fair point and those recent studies (among others) found that the footprint can be asymmetric. In our study as well, the rectangular shape of the field inevitably leads to minor deviations from a circular shape. This might be kept in mind, yet, as the overall study design focuses on symmetrically aligned topographical elements, such considerations are only of theoretical interest. Only in cases where there are entities of large soil moisture differences close to the sensor and heterogeneously positioned around the sensor, footprint deformations can be of interest. Also, as we stated later in the manuscript, the angular distribution is not subject of this study. We thus believe that the assumption of a circular footprint fits the scope and topic of this study and is the best choice.*

*Nonetheless, to provide the reader with additional information, we decided to extend this topic and include the references proposed by the reviewer. The revised text reads: "Although some studies suggested an asymmetric or "amoeba-like" footprint (Schattan et al. 2019; Schrön et al. 2022), most studies assume a circular footprint that depends on the Euclidean distance between the points where neutrons had first contact with the ground and the point of detection."*

- Figure 1: Please indicate the detector position (e.g. with a point) and add scales (in meters) to the inner area box (hectares alone are not very easy to grasp)

*As mentioned in previous replies to the comments, the suggested changes were included in Figure 1 (see Figure R 3).*

- L194-197: These statements sound like results, rather than methods. Or are they already established knowledge? Then please provide references.

*We included in the text some references to previous works of Weimar et al. (2020) and Köhli et al (2018).*

- L199: Consider adding Rasche et al. 2021, doi:10.5194/hess-25-6547-2021, as a reference for the discussion on thermal neutrons and their behaviour in heterogenous terrain.

*We included the suggested reference.*

- L201: Why "either"? Can't you use the same neutron for both, count rate calculation and footprint calcuation?

*The previous manuscript version was not fully clear here. The revised version reads: "Then, these weights were summed to obtain the number of detected neutrons and subsequently used in a weighted calculation of the R86".*

- L214: As you have mentioned also on other places, consider adding Schrön et al. 2022, as they seem to have demonstrated exactly that.

*We included the suggested reference.*

- L215: "Here, it is assumed that having a relatively small R86 is beneficial when monitoring irrigation in small fields" - Please rephrase, what does this assumption imply?

*The previous manuscript version was not fully clear here. The revised version now reads: "Here, the initial hypothesis is that a relatively small R86 is beneficial when monitoring irrigation in small fields as a lower contribution from the surrounding areas could be expected".*

- Eq 1: Please elaborate more on where this equation comes from. It looks like you are propagating the error of N1-N2 (which is the change of neutrons upon detection), where the error of Ni is 1/sqrt(Ni)?

*The reviewer correctly interprets Eq. 1. This is the result of Gaussian error propagation on ND = N1-N2, which means taking the square root of the sum of the squared derivatives of ND times its error which is, due to counting statistics, the square root of the resulting number of counts in N1 or N2, respectively. We did not intend to add a long description about basic statistics and thus provided an easier reading. We believe that the original text is sufficient in this case.*

- L232: The cited preprint is not a good reference here. Can you refer to a study which presents typical irrigation intervals?

*We understand the point of the reviewer, as the original text was misleading. We could not find relevant information on detection uncertainty for generic CRNS in evenly irrigated small fields. Thus, we included a generic detection uncertainty for soil moisture monitoring. The revised version of the manuscript clarifies this aspect and reads: "In addition to σ, a value of α=1% was included in each threshold to represent a generic detection uncertainty limit for a detector that can achieve ~1000 counts per hour and aggregation times of < 12 hours that are relevant in SM monitoring (Schrön et al., 2022)".*

- L248: "inimum and maximum percentages of detected neutrons" - Is this relative to all albedo neutrons, or to all detected neutrons (including non-albedo neutrons)? Please clarify.

*We carefully considered the comment of the reviewer here. We now provide a clear description, and the revised text reads: "For each dimension of the simulated inner area and for each moderator type, Table 2 shows the minimum and maximum percentages of detected (albedo plus non-albedo) neutrons that originate in the inner area depending on SM conditions".*

- Figure 5: Since panel f) shows a difference rather than the actual value of R86, the colorscale should be completely different. However, it still has blue and yellow colors just like the colormap from the other panels. Please make f) more distinguishable, by choosing a more distinguishable colormap (e.g., red-white-black). Moreover, using terrain colormaps for R86 is not a good choice. Blue stands for water, but low radii have nothing to do with water. Try to use a more linear colormap (e.g., greyscales), or none.

*We carefully considered the comment of the reviewer and critically examined Figure 5 of the original manuscript (Figure 4 of the revised manuscript). We agree that a simpler and clearer version could be produced. As low R86 can be somehow associated with wet soil conditions and large R86 with dry soil conditions, we decided to use a simpler blue-red scale for the panels a-e. Thus, green (intensity) was a good contrasting choice for panel f. We tested the use of greyscales, but it made the black contours less readable. Also, the use of no colormap resulted in a figure that was like Figure 3 of the original manuscript, which we now understand had poor readability. Thus, we believe that this revised Figure 4 (see* Figure R 7*) is a better and readable compromise. We also now include both a 5 mm HDPE moderator and a 25 mm HDPE moderator with gadolinium shielding. This because they show the smallest and largest footprints. In order to include both moderators without making the figure too complex, we excluded results from irrigated areas of 1 ha, 2 ha, and 4 ha as these do not show any special or additional feature worth mentioning at this stage.*

[Figure]

*Figure R 7: alternative figure that substitutes Figure 5 of the original manuscript (Figure 4 of the revised manuscript).*

- Figure 6: Can you add conventional N(sm) functions for the purely homogenous case for comparison?

*Following the reviewer suggestion, we included relative change in detected neutrons for homogeneous SM variations (i.e., homogeneous area or irrigated field of infinite dimension). The result is shown in the following* Figure R 8. *As this revised Figure 6 is more informative than the previous version, it was added to the revised version of the manuscript. The caption was modified accordingly.*

[Figure]

*Figure R 8: alternative figure that substitutes Figure 6 in the revised version of the manuscript.*

- Figure 8: Why are some bars larger for higher sm than others at lower sm? Is this an effect of the simulation uncertainty? Can you provide an errorbar for the bars?

*The results of this study are subject to statistical uncertainties. When looking at small differences, results are often influenced by fluctuations, which we for example see in Figure 8. In order to quantitatively understand and classify our findings, we provide a measure of certainty by comparing them to the standard deviation of the respective data set. In our opinion, the dashed lines show the error bars of the bars in a clearer and simpler manner. We thank the reviewer here as we also noticed that the red area below σ+α mentioned in the caption of Figure 8 was not visible. We also now use multiple simulation results for the initial homogeneous soil moisture conditions. We thus provided a revised version (Figure R 9) of the figure where such area is clearly marked, which will also help the reader in this context. We also modified part of the caption to have a clearer description: "The bars show the relative change in detected neutrons induced by the irrigation event while the dashed lines show the prescribed detection certainty thresholds". Please note that further modifications to Figure 8 were made according to the comments of reviewer n.2.*

[Figure]

*Figure R 9: alternative figure that substitutes Fig.8 in the revised version of the manuscript.*

- Figure B1: % relative to what? Can you add the signal of a completely bare sensor (0 mm HDPE)?

*The figure shows the percentage of detected thermal neutrons over total detected neutrons with different moderator types for the 1 ha scenario relative to the total number of neutrons detected. To make this clearer, we improved the caption, which now reads: "Boxplot of the percentage of detected thermal neutrons over the total number of neutrons that are detected with different moderator types for the 1 ha scenario". As the figure shows the fraction of thermal neutrons that are detected and not the count rate, the inclusion of a bare counter to the plot by means of additional investigations would not, in our opinion, provide meaningful additions to the manuscript. According to the general understanding, a thermal counter counts 100 % thermal neutrons. That is, however, not exactly true and the neutron absorption probability in a neutron*

*converter has a 1/sqrt(E) dependence. Given the threshold of thermal energies, the thermal counter would rather lie around 90 % of thermal neutrons. Starting a discussion about why that is the case is not the focus of this manuscript. Also, we believe that at this point of the manuscript and of the study, the inclusion of a 0 mm HDPE moderator would not yield any further understanding in the context of our research questions.*

- The reference list is sorted by first author name, but for the same author it is not sorted by publication year. This makes searching for references very cumbersome, especially for extensivley used author names (such as Bogena et al.). It should be fixed during typesetting.

*This was improved in the revised version of the manuscript.*

- With regard to the previous comment, please double-check whether such a high number of rather general references on soil moisture are necessary for this very specific manuscript about neutron detector simulations and irrigation.

*We removed the references of those parts of the introduction that were shortened according to previous comments.*

**Response to Anonymous Reviewer n.2**

This paper investigates the feasibility of CRNS-based SM monitoring in irrigated environments. The paper is informative with lots of simulations in different scenarios. Simulation of neutron count in different scenario was performed with Monte Carlo simulations and Ultra Rapid Adaptable Neutron-Only Simulation. However, obtained results were not validated using real case scenario of irrigated areas.

*We hope that our work can address all the points raised by the reviewer and that this revised version of the manuscript can be accepted for publication in Geoscientific Instrumentation, Methods and Data Systems.*

The author chose a square, not a circle, irrigated area. While the CRNS is tube-shaped and the footprint of the CRNS is circular, is there a reason for that specific shape? Also, do we expect an improved detection rate or minimal if he changed it to circular?

*We believe that a circular pivot irrigation field would have not been the best choice for this study because circular centre-pivot irrigated fields are very large and typically 400 m in radius (although 500 m systems are also common and larger systems exist). Small centre-pivot irrigated fields are not common. The large dimension of a centre-pivot irrigated field means that most of the neutrons detected by a CRNS placed in the middle of such large field (if not all detected neutrons) originate within the irrigated field. Thus, the irrigation that the CRNS would sense is comparable to a rain event, and we think this would not be as interesting as a smaller field of a few ha where the outside area plays a role.*

*Relatively small fields as those investigated in this study are most often rectangular and not circular. It is true that the rectangular shape can vary greatly and elongated rectangular shapes are common, but the inclusion of elongated shapes in this study would lead to a much more complex manuscript and would go beyond the scope of the manuscript. Since the manuscript is already long and complex in the current form, we believe that the reader would have strong difficulties in navigating through additional shapes of the irrigated area. Nonetheless, it could be expected that such elongated shapes would be more challenging for the CRNS compared to a square shape. We agree with the reviewer that the selection of the shape is an important topic, and we included additional considerations in Chapter 4 "Limitations and outlook". We also think that future research, especially in real-case scenarios, should investigate these aspects and we mentioned this in the revised manuscript.*

*To obtain first insights into the effects of field shape, we performed additional simulations of circular and rectangular (142x70 m) irrigated fields of 1 ha area and compared the results with those of a square 1 ha irrigated field. The results are shown in* Figure R 10 *in a similar way as they are shown in Figure 8 of the manuscript. In general, differences between the square, circular, and rectangular shape are rather small, at least in this simulation setup. Compared to the square shape, there might be a small tendency towards higher relative changes in detected neutrons for a circular shape and a tendency towards lower relative changes when a rectangular shape is used. However, results are too similar to draw meaningful conclusions. A clearer picture could possibly be obtained if these two additional scenarios are simulated for the entire soil moisture range of 0.05 to 0.50 $cm^3$ $cm^{-3}$ and for the five investigated areas of the irrigated field. However, this goes beyond the scope of the manuscript and would additionally increase the number of simulations by +200%, which is not feasible due to time and computational constraints. Thus, we decided not to add simulations of different field shapes to the manuscript. Nonetheless, considerations on the shape of the irrigated field were added to Chapter "4 Limitations and Outlook" and to the conclusions of the revised manuscript.*

[Figure]

*Figure R 10: CRNS chance of detecting irrigation events of 0.05 and 0.10 cm3 cm-3 (blue and green bars respectively) in a) squared, b) circular, and c) rectangular irrigated field of 1 ha. The bars show the relative change in detected neutrons induced by the irrigation event while the dashed lines show the prescribed detection thresholds. The red area below the σ+α threshold indicates uncertain detection.*

Below few questions that could make the paper clearer for the reader:

*We have carefully examined the comments provided by the reviewer and we offer here a point-by-point answer.*

Line 14: the unit needs typo correction

*The typo was corrected.*

Introduction: Recent work on soil moisture mapping from SAR images is worth reporting in the introduction, primarily those that provide operational soil moisture mapping through the synergistic use of Sentinel-1 and Sentinel-2.

*We agree with the reviewer that such studies on radar-derived SM products are of general interest and could fit the first paragraphs of the introduction. However, reviewer n.1 pointed at the length of the introduction, which was shortened in the revised version of the manuscript, and at the large number of general references on soil moisture. We thus preferred not to add these further references to the manuscript.*

In line 143-144, you mentioned that variations in humidity, vegetation, and other environmental variables can affect the footprint but with less degree than the SM effect. However, in the simulation, these factors were fixed (line 167-170). Can you explain what the impact would be on the simulation results if you include the diurnal weather conditions?

*We can expect that the diurnal changes in humidity and other environmental variables will affect the CRNS footprint and count rate as mentioned in the manuscript. Regarding the count rate, correction procedures exist for most of such variables and in real-world applications. For example, although atmospheric humidity could vary in an irrigated field, this is typically measured, and the count rate is corrected accordingly.*

*The effect on the footprint, on the other hand, cannot currently be corrected but only explored using neutron transport simulations. Based on previous simulations studies, we can expect a variation in the footprint due to atmospheric humidity changes as shown by Köhli et al. 2015. However, the investigation of a second humidity value would double the quantity of simulations and results. As multiple air humidity values would need to be simulated to obtain meaningful results, we believe that this would result in a too complex picture for the reader and, overall, in a confusing and unfocused manuscript. The same applies to other variables such as vegetation and this would go beyond the scope of the manuscript as the focus is on soil moisture. Nonetheless, we agree with the reviewer that such effects should be mentioned and possibly explored in future research, and we added considerations on this matter to Chapter "4 Limitations and outlook".*

Line 187: why did you choose 9m radius specifically for all the simulations? Is there a method for choosing the right radius size for the tube?

*The dimension of the virtual detector was set to 9 m as this is commonly done in such simulations with the URANOS model. In selecting the dimension, two aspects should be considered:*
   *a) The smaller the virtual detector, the lower the chance of detecting a simulated neutron and thus the lower the statistical significance of the simulation. This can be counterbalanced by a higher number of simulated neutrons, which however can considerably extend the simulation time and computational needs.*
   *b) A larger virtual detector has higher chance of detection and thus higher statistical significance of the simulations. But the more the dimension of the virtual detector is stretched beyond that of the actual detector, the more secondary effects can influence the simulation results.*

*In the end, a 9 m radius (generally below 5% of the footprint) is a good geometrical limit (and a good compromise between the above-mentioned aspects). This is valid for a typical analysis where there are no variations of the environmental topology in the immediate vicinities of the virtual detector. As these considerations are discussed in some of the literature that the manuscript refers to, we believe that there is no need at this point for additional information on such detail of the methods.*

Looking at Table2 + Figure 3 and then Figure 6, Figure 7 and Fig 9. They are very well connected, I wonder why they did not come directly after each other so that it is easier for the reader to stay on track!

*We agree with the reviewer that such alternative order of the results can offer a nice storyline and a clearer reading. In the revised version of the manuscript, we reorganized the results Chapter. We first discuss the footprint dimensions (note that previous Figure 4 and Figure 5 are now Figure 3 and Figure 4, respectively) and then the other aspects (Table 2, Figure 3 (now Figure 5), Figure 6, Figure 7, Figure 8, and Figure 9). We adapted the consistency of other Chapters such as introduction, and conclusions. Such changes did not alter the overall results or the message of the manuscript but should result in a better reading experience.*

Lines 374 – 376: "As shown in Figure 8, an irrigation event that leads to a 0.05 cm3cm-3 increase in SM can be detected with CRNS (relative change in detected neutrons higher than $3\sigma + \alpha$) when the initial SM of the simulation domain is 0.05 cm3cm-3." It looks like there is an overlap in this figure: the green bars represents the 0.05 cm3cm-3 initial SM and the blue bar for the 0.10 cm3 cm-3 . Indeed, the relative change is lower when the increase of soil moisture is higher. Is that correct?

*We thank the reviewer for the comment as we now understand that the readability of Figure 8 should be improved. In Figure 8, the blue bars on the left plots refer to a 0.05 $cm^3$ $cm^{-3}$ irrigation event and the green bars to a 0.10 $cm^3$ $cm^{-3}$ event. The initial soil moisture is indicated on the X-axis at the bottom of the figure. Thus, the relative change is higher when the increase in soil moisture due to irrigation is 0.10 $cm^3$ $cm^{-3}$ (green bars, left side). The relative change is generally lower when the initial soil moisture is higher.*

*To improve the readability of the figure, we included titles with the irrigation amount on top of the X-axis and we further improved the legend. The revised figure clearly refers to irrigation-related soil moisture changes (see following* Figure R 11*). We believe that this revised version is a meaningful improvement, and we thank the reviewer for his comment. Please note that further modifications to Figure 8 were made according to the comments of Reviewer n.1 and due to the current use of multiple simulation results for the homogeneous initial soil moisture conditions.*

[Figure]

*Figure R 11: alternative figure that substitutes Figure 8 in the revised version of the manuscript.*

---

## Author Response (AR2)

**Response to reviewers**

*Please find here our response to the remaining reviewer comments. In our opinion, all concerns raised by the reviewer were addressed and this has further strengthened the message and variety of the manuscript. We hope that this revised version of the manuscript can be accepted for publication in Geoscientific Instrumentation, Methods and Data Systems.*

**Response to Reviewer 1**

Thank you very much for carefully considering all review comments and for comprehensively elaborating on all raised concerns. I agree to most of them, while I think that few concerns are still not sufficiently met and require minor additions.

*We believe we have addressed the additional concerns. Please find below a point-by-point reply.*

My major concerns 2 and 3, all minor concerns, and all specific comments were sufficiently addressed, thank you.

Major concern 1: The authors defended their choice of a constant soil moisture profile from 0 to 160 cm with good arguments ("typical" profile shapes, near-field sensitivity, high complexity). However, the added text to the revised manuscript does not sufficiently address the limitations of this study in a useful way. When strong conclusions about the feasibility of irrigation detection are made, then the underlying assumptions and limitations should be strongly communicated as well! Here, I would expect more than two sentences, and a better quantification than "reduction of performance". The readership should clearly understand that these results are only valid for a very specific set of soil properties and vertical profiles. Otherwise, I see the risk that readers may misinterpret and unintentionally miscommunicate the results (e.g., by comparing them with observations and finding weak agreement). Without opening a full new dimension of complexity, I think a rough quantitative statement about the ranges and uncertainties could be achieved by a few quick simulations of a single scenario (e.g., 1 ha and 25mmG, theta=0.01, dtheta=0.20) with irrigation of the first 10 cm instead of 160 cm. My hypothesis would be that this more realistic soil moisture profile would lead to much less impact of the outer region. By a quick comparison the 10 cm and the 160 cm results you would be able to either support your conclusion (soil profile has low impact) or quantify the limitations (impact may vary by X % depending on soil profile). Moreover, the conclusion which shield is the best may depend on the vertical soil profile, while it might lead to very impactful economical decisions for manufacturers and customers in the future. So, also here the authors should take the responsibility by making a few tests with more realistic soil profiles to quantify the robustness of the 25mmG performance. Also here, a rough estimate from just a few sample simulations would be better than no estimate.

*We understand the concern of the reviewer regarding the brevity of the limitations and outlook section. This section discusses a wide range of possible limitations and all of these, although briefly described for the sake of conciseness, are clearly pointed out. But we also understand the need for a quantification of this specific limitation so that readers that are not familiar with the methods can have a better overall understanding of the applicability of the results.*

*In our opinion, a comprehensive quantification of the impact of vertical soil moisture heterogeneity and of the wetting of the upper soil would necessitate a) the definition of one or better multiple exemplary vertical SM heterogeneity patterns, b) large number of additional simulations, and c) a considerable addition to the "Results and Discussion" section. We believe that such a comprehensive quantification goes beyond the scope of the study and is sufficient material for a completely new study. We also think that real-world data would be useful to support such a new study.*

*Thus, we followed the reviewer's suggestion to include a brief quantification of the possible reduction in CRNS performance due to the wetting of the topsoil. For this, the new version of the manuscript includes a new Fig. 10 (see Figure R 1 below) and relative descriptions in Chapter 4 "Limitations and outlook". This new text also includes elements that were added in response to Major Concern 5. The overall content of this new text and responses to the reviewer's concerns and hypothesis are described in detail in the following, whereas the new text in the manuscript is briefer on certain points for the sake of conciseness.*

*We tested the scenario proposed by the reviewer with one simulation that had a SM of 0.01 $cm^3$ $cm^{-3}$ and a second simulation where the SM of a 1 ha irrigated field was increased to 0.20 $cm^3$ $cm^{-3}$ only in the upper 10 cm of soil (thus, the SM in the non-irrigated area and in the irrigated soil below 10 cm remained at 0.01 $cm^3$ $cm^{-3}$). With a 25 mm HDPE moderator with gadolinium shielding, the number of detected neutrons dropped by 26.2 %. This reduction easily satisfies the thresholds that were used in the manuscript and is stronger than any of the scenarios proposed in Fig. 8 of the manuscript (the strongest variation for a 1 ha scenario being ~19 % with a SM increase of 10 $cm^3$ $cm^{-3}$ from a starting SM of 0.05 $cm^3$ $cm^{-3}$). We believe this is due to the relatively low initial SM and to a relatively large SM variation. Consequently, we decided to not use this scenario and instead added additional simulations to address the concerns. The newly added simulations are consistent with the simulations presented in Fig. 8 of the original manuscript to achieve consistent text flow and to minimize the complexity of the newly added text. The new results are shown in Figure R 1 (panels a and d-e).*

*In these new simulations, we considered two scenarios that were applied to the 1 ha case:*
1. *A wetting of only the first 10 cm of soil with a SM variation of 0.05 and 0.10 $cm^3$ $cm^{-3}$. This could correspond to the initial wetting that takes place after irrigation starts. In fact, such SM variation equals to only 5 mm (0.05 $cm^3$ $cm^{-3}$ variation) and 10 mm (0.10 $cm^3$ $cm^{-3}$ variation) of irrigation that penetrates the soil, which is generally a rather small amount.*
2. *A wetting of the first 30 cm of soil with a SM variation of 0.05 and 0.10 $cm^3$ $cm^{-3}$. This could correspond to a later stage of irrigation when, for example, the water has moved through the plough layer. We believe that this new depth is suitable and still conservative as the roots can reach deeper soils. These SM variations are equal to 15 mm (0.05 $cm^3$ $cm^{-3}$ variation) and 30 mm (0.10 $cm^3$ $cm^{-3}$ variation) of irrigation. Although larger than in scenario 1, such irrigation amounts are consistent with real-world irrigation practices.*

*These new results did not clearly provide new proofs of a possible reduction of the impact of the other region. We could not calculate the impact directly from our simulations, as we only modified the SM in the irrigated area. Such calculation would require additional simulations. However, we analysed the partitioning between neutrons that originate inside or outside the irrigated field. The*

*partitioning obtained with the new simulations (10 cm and 30 cm wetting) was rather similar to that of the previous simulations with homogeneous vertical SM. Only minor differences of ±2 % could be observed and the percentage of detected neutrons that originate in the outer region was not consistently higher or lower in these new simulations. One could think that changing the SM of the outer region only in the topsoil could lead to a lower influence of such region (because the overall SM change is smaller). However, the reduction of the depth of investigation of a CRNS with distance should be considered here. Also, such influence would strongly depend on the SM difference between topsoil and deeper soil, both within and outside the irrigated area. All in all:*

- *we do not think we can either support or clearly reject the hypothesis of the reviewer, even with the additional simulations that were performed.*
- *we do not think that we have sufficient material to discuss the influence of the outer region for this specific case*
- *and we believe that the assessment of a possible variation in the influence of the outer region due to vertical SM heterogeneity goes beyond the scope of the current study.*

*Thus, we did not mention these aspects in the revised text.*

*We compared the use of a 25 mm HDPE moderator with Gadolinium shielding to that of a 25 mm and of a 5 mm HDPE moderator without shielding in the context of Fig.10 of the revised manuscript (i.e., 1 ha field). In general, the performance of the unshielded versions was poorer than that of the gadolinium shielded version. In particular, the 25 mm HDPE moderator had performance reductions up to -29 % (this for 0.05 cm$^3$ cm$^{-3}$ irrigation starting from SM = 0.05 cm$^3$ cm$^{-3}$). Such reduction was smaller when the initial SM was higher. When a 5 mm HDPE moderator was used, reductions in performance exceeded -90 %. Interestingly, when only the first 10 or 30 cm of soil were wetted, the performance of the unshielded versions was reduced even further (e.g., up to -55 % for 0.05 cm$^3$ cm$^{-3}$ SM variation and initial SM = 0.05 cm$^3$ cm$^{-3}$, 25 mm HDPE moderator). Although these results are in principle interesting, we believe that such sensitivity differences are already discussed in detail in section 3.3 and in Fig. 6 of the manuscript). These new results do not change the findings of these sections. It is true that there is a general additional point in favour of the sensitivity of the gadolinium shielding version, but we believe that these new simulations are not sufficient to draw definitive conclusions. Thus, we did not include new parts on the best performing moderator-shielding in the new text of Chapter 4.*

*The reduction in performance of the CRNS when only the first 10 cm or 30 cm of soil are wetted is shown in Figure R 1 (and Fig.10 of the revised text). It is clearly shown (panel d) how there is a strong reduction of performance for a wetting of the first 10 cm and a SM variation of 0.05 cm$^3$ cm$^{-3}$ (blue bars in Figure R 1d). The performance reduction is less pronounced with a SM variation of 0.10 cm$^3$ cm$^{-3}$ (green bars in Figure R 1d). Even though this reduction in performance is apparent, we do not believe that this results in the unsuitability of CRNS for irrigation monitoring or in a strong undermining of our previous results. As pointed out earlier, such irrigation events are limited to 5 mm and 10 mm of water that penetrates the soil, which is a rather small amount and could correspond to the initial phase of a larger irrigation event (or daily irrigation events). Thus, we emphasized in the revised text that CRNS may prove less suited to monitor initial irrigation stages of a larger irrigation event as well as daily small irrigation events.*

*In the case of 30 cm of wetted soil (Figure R 1e), the results are rather similar to those of a homogeneous vertical wetting (see red dashed bars). In general, a small reduction in performance*

*could be observed at times, but the results show an overall good performance in this scenario that is comparable to that of a homogeneous vertical SM distribution. We believe that these results are a consequence of the CRNS sensitivity that peaks in the topsoil (especially when the topsoil is wetted) and of the fact that the CRNS depth of penetration decreases far from the instrument.*

[Figure]

*Figure R 1: new Fig.10 of the revised manuscript.*

*A full interpretation of these results would require additional simulations, but we believe that they offer an interesting picture. Although not exhaustive, we believe that they are a nice and informative addition to Chapter 4 "Limitations and outlook", but we emphasized in the revised text that further and more detailed simulations would be needed to better understand such effects.*

*The revised text, which includes* Figure R 1 *as the new Fig.10 together with additions from major concern 5, takes this into account and reads:*

*"Moreover, heterogeneous vertical SM distributions or different dimensions of the irrigated field, such as irregular or elongated shapes, might be more challenging for irrigation monitoring with CRNS. Figure 10 shows the sensitivity of a CRNS with a 25 mm HDPE moderator and gadolinium shielding to irrigation events that increase the SM of the irrigated area by 0.05 or by 0.10 $cm^3$ $cm^{-3}$ starting from a homogeneous SM condition for different shapes of the irrigated area. The comparison between a circular (56 m radius), rectangular (142x70 m), and a square field of 1 ha (Figure 10a-c) shows that there is a small change in CRNS performance for given SM variations for different field geometries. However, the differences are small and the overall feasibility of irrigation monitoring with CRNS is not affected. Figure 10 also shows the sensitivity to irrigation events when only the first 10 cm or 30 cm of soil are wetted in a 1 ha square field. When irrigation affects SM only in the first 10 cm of soil (Figure 10d), the sensitivity of the CRNS is strongly reduced. This is especially the case for SM variations of 0.05 $cm^3$ $cm^{-3}$ where the CRNS is able to detect irrigation only when the initial SM is 0.05 $cm^3$ $cm^{-3}$. However, it should be noted that SM variations of 0.05 and 0.10 $cm^3$ $cm^{-3}$ correspond to irrigation events of 5 and 10 mm. This rather small irrigation amount might correspond to the initial SM variation during a larger irrigation event or to frequent events (e.g., daily irrigation). When irrigation affects SM in the first 30 cm of soil, the sensitivity of a CRNS is comparable to that of a homogeneous vertical distribution of SM (Figure 10a and e). The only differences are a drop from high to good detection chances for a 0.05 $cm^3$ $cm^{-3}$ SM variation and from certain to high detection chances for a 0.10 $cm^3$ $cm^{-3}$ SM variation when the initial SM is 0.15 $cm^3$ $cm^{-3}$. Overall, Figure 10 suggests that a small change in the field shape and irrigation affecting just the top 30 cm of soil will only have a small influence on the feasibility of irrigation monitoring with CRNS. However, it has to be noted that these results are based only on a limited number of simulations and real-world studies should assess in more detail the influence of the shape of the irrigated field in addition to the impact of the within-field SM heterogeneity."*

*The conclusions were also modified with the addition of the following text:*

*"For a 1 ha irrigated area, the use of a circular or rectangular shape instead of a square shape and SM increases in the first 30 cm of soil instead of the entire soil profile did not results in considerable changes in the CRNS sensitivity. On the contrary, when SM was increased only in the first 10 cm of soil (e.g., small daily events or the initial stage of larger irrigation event), a considerable reduction in sensitivity was observed."*

*Finally, Chapter 2.4 "Simulation Setup" and Chapter 2.6 "Investigation of the feasibility of irrigation monitoring with CRNS" were extended to include the simulation of these new scenarios.*

Major concern 4: I am okay with the added explanation that a single soil moisture sensor may be installed outside the irrigated field to support the CRNS *if* the outside field is homogeneous. I just want to remind the authors that, if it is not homogeneous (probably in most cases), it would be more logical and less expensive to use the single soil moisture sensor in the irrigated field, and no CRNS. Since the inner field is always much smaller than the outer field, the single soil moisture sensor will be much more representative for the inner field than for the outer field. Hence, the

overall representativity-related uncertainty introduced by your suggested solution might be much higher compared to the inner single SM sensor without CRNS. You might want to consider communicating this dilemma, although it does not support the recommendation of CRNS for small, irrigated fields.

*We understand the concern of the reviewer here and agree that a crystal-clear text should be obtained to address this important point. However, we do not believe that installing a single point-scale SM sensor in the irrigated field is always the best option (although it is the cheapest one). The reviewer rightly points at the fact that the SM outside a target irrigated field is, in most cases, spatially heterogeneous. However, we think that it is rare to find homogeneous SM distributions in agricultural fields when these are surrounded by lands with heterogeneous SM distributions. Most agricultural fields will also have a heterogeneous SM distribution. When the irrigated field is very small, however, it has higher chances of showing a relatively homogeneous SM distribution. Nonetheless, in the case of a within-field heterogeneous SM distribution, a single sensor is of little use and sensor networks are generally necessary to avoid the risk that a single sensor is placed in a non-representative position and induces a considerable bias compared to the field average. Thus, CRNS are proposed as a method to get a field measurement with a single sensor, as mentioned in the introduction.*

*We understand that we did not communicate this message in a complete manner in Chapter 4. Also, we did not express sufficiently the comparison of a CRNS with sensor networks or single instruments in the case of small, irrigated fields. The revised text takes these considerations into account and reads:*

*"Another aspect that needs further investigation is the role of SM variations in the surroundings of a target irrigated field. Information on such SM variations may be necessary to correct CRNS-based SM products not only in relatively small fields (up to 2 ha) but also in larger ones. When a CRNS is installed in an irrigated field in place of a sensor network, the CRNS could be supported by a single and inexpensive point-scale SM monitoring instrument installed outside the target field. This would not substantially increase the installation and maintenance costs and would not interfere with agricultural management. Moreover, a single point-scale device could support multiple CRNS in an agricultural area if irrigated fields are sufficiently distanced and if the SM in the unmanaged area is relatively homogeneous in space. However, in small fields (e.g., < 0.5 ha) that have relatively homogeneous SM, a small sensor network could be more effective than a CRNS.".*

Major concern 5: Thank you for the additional simulations. I recommend using these results to add a concrete number to the discussion: calculate the average deviation of circular and rectangular results from the square results. Then you can conclude that the shape of the field impacts the results by X %. Then, it is easier for the reader to decide about the transferability of your results to individual sites.

*We believe that, given the results shown in Figure R 1, the best option is to include such results in the text of Chapter 4: "Limitations and outlook" together with the revisions of major concern 1. We tried to calculate an average value for the 3 scenarios proposed in such figure. These would*

*be a variation of 8.9 %, 8.7 %, and 8.5 % in scenarios a) 1 ha square, b) 1 ha circular, and c) 1ha rectangular field. These values are rather similar and are a strong simplification of the results of Figure R 1, which shows that the sensitivity depends on the initial SM conditions. Thus, the option to show the results in a new Fig.10 of the revised text is, in our opinion, the most appropriate choice. The revised Chapter 4 and conclusions (see comments to major concern 1) include this new figure as well as a description of the simulated scenarios and of the fact that only small differences were found when using a circular or rectangular field shape instead of a 1 ha squared shape. The new text also points at the fact that more studies and analysis are needed to get a complete picture of these effects as these few simulations can be considered only an indication and are not conclusive results. Chapter 2.4 "Simulation setup" was extended to include the simulation of these new scenarios.*

Other comments: The abstract reads "However, it was found that variations in SM outside a small, irrigated field (i.e., 0.5 and 1 ha) can affect the count rate more than SM variations due to irrigation." - Consider rephrasing to clarify that this is only true under some (and by no means all) conditions.

*The revised text now reads: "However, variations in SM outside a 0.5 or 1 ha irrigated field (e.g., due to irrigation of neighbouring fields) can affect the count rate more than SM variations due to irrigation."*